# Cézeaux-Aulnat-Opme-Puy De Dôme: a multi-site for the long term survey of the tropospheric composition and climate change

Jean-Luc Baray[1,2], Laurent Deguillaume[1,2], Aurélie Colomb[1], Karine Sellegri[1], Evelyn Freney[1], Clémence Rose[1], Joël Van Baelen[1,3], Jean-Marc Pichon[1,2], David Picard[1], Patrick Fréville[2], Laetitia Bouvier[1,2], Mickaël Ribeiro[1], Pierre Amato[4], Sandra Banson[1], Angelica Bianco[1,4], Agnès Borbon[1], Laureline Bourcier[1], Yannick Bras[1], Marcello Brigante[4], Philippe Cacault[2], Aurélien Chauvigné[1,5], Tiffany Charbouillot[1,4], Nadine Chaumerliac[1], Anne Marie Delort[4], Marc Delmotte[6], Régis Dupuy[1], Antoine Farah[1], Guy Febvre[1], Andrea Flossmann[1], Christophe Gourbeyre[1], Claude Hervier[2], Maxime Hervo[1,7], Nathalie Huret[1,2], Muriel Joly[1,4], Victor Kazan[6], Morgan Lopez[6], Gilles Mailhot[2,4], Angela Marinoni[1,8], Olivier Masson[9], Nadège Montoux[1], Marius Parazols[1,4], Frédéric Peyrin[2], Yves Pointin[1], Michel Ramonet[6], Manon Rocco[1], Martine Sancelme[4], Stéphane Sauvage[10], Martina Schmidt[6,11], Emmanuel Tison[10], Mickaël Vaïtilingom[1,4,12], Paolo Villani[1], Miao Wang[1], Camille Yver-Kwok[6], Paolo Laj[8,13]

[1] Université Clermont Auvergne, CNRS, Laboratoire de Météorologie Physique, UMR 6016, Clermont Ferrand, France
[2] Université Clermont Auvergne, CNRS, Observatoire de Physique du Globe de Clermont Ferrand, UMS 833, Clermont Ferrand, France
[3] Université de la Réunion, CNRS, Météo-France, Laboratoire de l'Atmosphère et des Cyclones, UMR 8105, St Denis de la Réunion, France
[4] Université Clermont Auvergne, CNRS, SIGMA Clermont, Institut de Chimie de Clermont-Ferrand, UMR 6296, Clermont-Ferrand, France
[5] Université Lille, CNRS, Laboratoire d'Optique Atmosphérique, UMR 8518, Villeneuve d'Ascq, France
[6] Université Paris-Saclay CEA-CNRS-UVSQ, Laboratoire des Sciences du Climat et de l'Environnement, UMR 8212, Gif-sur-Yvette, France
[7] MeteoSwiss, Payerne, Switzerland
[8] Institute for Atmospheric Sciences and Climate, National Research Council, Bologna, Italy
[9] Institut de Radioprotection et de Sûreté Nucléaire, Saint Paul lez Durance, France
[10] IMT Lille Douai, Univ. Lille, SAGE – Département Sciences de l'Atmosphère et Génie de l'Environnement, Lille, France
[11] Institut für Umweltphysik, University of Heidelberg, Heidelberg, Germany
[12] Université des Antilles, Laboratoire de Recherche en Géosciences et Energies, EA 4539, Pointe-à-Pitre, Guadeloupe, France
[13] Université Grenoble-Alpes, CNRS, IRD, INPG, IGE UMR 5001, Grenoble, France

*Correspondence to*: Jean-Luc Baray (J.L.Baray@opgc.fr) and Laurent Deguillaume (laurent.deguillaume@uca.fr)

**Abstract.** For the last twenty-five years, CO-PDD (Cézeaux-Aulnat-Opme-puy de Dôme) has evolved to become a full instrumented platform for atmospheric research. It has received the credentials as national observing platform in France and internationally recognized as a global station in the GAW network (Global Atmosphere Watch). It is a reference site of the European and national research infrastructures ACTRIS (Aerosol Cloud and Trace gases Research Infrastructure) and ICOS (Integrated Carbon Observing System). The site located on-top of the puy de Dôme mountain (1465 m a.s.l.) is completed by additional sites located at lower altitudes and adding the vertical dimension to the atmospheric observations: Opme (660 m

a.s.l.), Cézeaux (410 m) and Aulnat (330 m). The integration of different sites offers a unique combination of *in-situ* and remote sensing measurements capturing and documenting the variability of particulate and gaseous atmospheric composition, but also the optical, biochemical and physical properties of aerosol particles, clouds and precipitations. Given its location far away from any major emission sources, its altitude and the mountain orography, the puy de Dôme station is ideally located to sample different air masses in the boundary layer or in the free troposphere depending on time of day and seasons. It is also an ideal place to study cloud properties with frequent presence of clouds at the top in autumn and winter. As a result of the natural conditions prevailing at the site and of the very exhaustive instrumental deployment, scientific studies at puy de Dôme strongly contribute to improving the knowledge in atmospheric sciences including the characterization of trends and variability, the understanding of complex and interconnected processes (microphysical, chemical, biological, chemical and dynamical) and the provision of reference information for climate/chemistry models. In this context, CO-PDD is a pilot site to conduct instrumental development inside its wind tunnel for testing liquid/ice cloud probes in natural conditions, or *in-situ* systems to collect aerosol and cloud. This paper reviews 25 years (1995-2020) of atmospheric observation at the station, and related scientific research contributing to atmospheric and climate science.

**1 Introduction**

Atmospheric greenhouse and reactive gases, aerosol particles and atmospheric water (water vapor, clouds and precipitation), as well as their interactions, play a crucial role for weather and climate systems (IPCC, 2013). Despite the vast improvement of global models and satellite observations, the necessity of ground based long term observations remains for model assimilation and validation, and allows the understanding of atmospheric processes to be improved. It is therefore crucial to maintain fully-instrumented stations for the long-term monitoring of the atmospheric composition with a high technological and scientific level. These stations are especially useful within the framework of larger observation networks such as the Global Atmosphere Watch program (GAW).

CO-PDD combines *in-situ* and remote sensing observations from different connected sites (Cézeaux-Aulnat, Opme and puy de Dôme) at different altitudes allowing the evolution of the composition of the troposphere of central France to be documented (Figure 1). The puy de Dôme station (PUY, 45.77°N, 2.96°E, 1465m a.s.l.) is located about fifteen kilometers far from immediate pollution sources, Opme (45.71°N, 3.09°E, 660 m) is located in a semi-rural area, and the Cézeaux (45.76°N, 3.11°E, 410 m) and Aulnat sites (45.79°N, 3.15°E, 330 m) are located in a suburban area, near the city of Clermont-Ferrand. The sites are 10 to 15 km apart.

PUY is the highest point of the "Chaîne des Puys", a north-south oriented volcanic mountain range, presenting orographic barrier to the prevailing oceanic westerly winds, documented by trajectory analysis (Hervo et al., 2014). The height of the mixing layer varies between day and night, and as a function of the meteorological conditions between 700 m and 2200 m a.s.l. (Farah et al., 2018). The altitude puts PUY below or above the atmospheric mixing layer, depending on season. PUY is in free tropospheric conditions 50% of the time in winter, but it is in the mixing layer most of the time in summer. In

addition, dynamical exchanges between these two atmospheric compartments can occur and significantly influence the lifetime and transport of aerosol particles injected in the free troposphere (Freney, 2016). A systematic analysis of webcam images shows that the PUY summit is in cloudy conditions on average 30% of the time, the cloud occurrence varies from 24% in summer to 60% in winter (Baray et al., 2019). The variety of cloud situations encountered (orographic, frontal, convective clouds...) makes the PUY station a strategic location for cloud observations.

The station serves the objective to provide to the scientific community a facility for hosting experiments and a set of long-term measurements to investigate the processes linking gases, aerosols, clouds and precipitation and the impact of anthropogenic changes on climate (cloud, radiation) and meteorology (precipitation). It is a unique place to study key research topics such as the nucleation of new particle and their interactions with atmospheric water vapor, bio-physico-chemistry of clouds or secondary organic aerosol formation resulting from gas-phase precursors.

CO-PDD has been in operation for more than 25 years. The purpose of this paper is to provide an overview of this station, including its history (Section 2), the scientific context and main scientific questions (Section 3), the national and international structuration (Section 4), a technical description of the observation systems (Section 5), some highlights of major scientific results derived from CO-PDD observations (Section 6), future plans and concluding remarks (Section 7). The data availability, the list of acronyms and the encountered air masses are given in Appendix A, B and C.

## 2 History

The puy de Dôme mountain has a very long history contributing to atmospheric sciences. Already, in September 1648, Blaise Pascal and his brother-in-law Florin Périer performed the Torricelli experiment at the puy de Dôme to prove the change of the atmospheric pressure with altitude. Two centuries later, the destruction of the French fleet during the Crimean War by a storm in 1854, was the triggering event for the study of meteorology in France. The history of the observatory at the top of puy de Dôme begins in 1869, when Émile Alluard (1815-1908), Professor of physics at the Faculty of Sciences of Clermont-Ferrand, proposed to the political authorities to install a mountain meteorological observatory at the top of the puy de Dôme. The construction began in 1872, was completed in 1875 and inaugurated on 22 August 1876. The observatory was composed of two stations: the upper station at the top of puy de Dôme summit, and a lowland station located about 1000 meters below, in the city of Clermont-Ferrand (Figure 2). The upper station was a seven meters tower, connected to the house of the station keeper by a tunnel, and equipped with thermometers, anemometers, mercury recording barometer, Hasler thermo-hygrograph, rain gauge, and condensation hygrometer (Anon, 1876). The responsibility of the observatory changed in 1925, becoming the Institut et Observatoire de Physique du Globe (IOPG) of Clermont-Ferrand. 45 years later, a research laboratory, the Laboratoire de Dynamique et Microphysique de l'Atmosphère (LDMA), was created at the University of Clermont-Ferrand by Professor Guy Soulage in 1970, becoming the current Laboratoire de Météorologie Physique (LaMP) in 1977 when associating with the French National Centre for Scientific Research (CNRS). In 1986, the Observatory merged with the Atmospheric Research Group and the Department of Earth Sciences to form an Observatory of

the Sciences of the Universe (OSU) named Observatoire de Physique du Globe de Clermont-Ferrand (OPGC). In this framework, starting in the late 80's, the LaMP/OPGC began to perform precipitation measurements with mobile instruments deployed in the basin of Clermont (Cézeaux and Opme) and dynamical measurements with wind profilers. Measurement campaigns have also been performed on other sites such as "Cévénol" region (ANATOL RADAR in S-Band, rain gauges and since 2006, disdrometers, micro rain and X-band RADARs). Cézeaux hosts also a LIDAR system in operation since 2008. Two intermediate sites were also instrumented for a better description of the dynamical and meteorological context: Opme in 1999 and more recently Aulnat in 2012. The microphysical and chemical observations at the puy de Dôme began in 1995. The puy de Dôme chalet was completely renovated in 2010. More than 70 atmospheric variables are currently measured at PUY, Cézeaux, Aulnat and Opme (Figure 3). The exhaustive list of measurement systems currently in operation at PUY, Opme and Cézeaux/Aulnat is given in Tables 1, 2 and 3 respectively. Figure 4 presents a time series of temperature measured at PUY, compiling historical and current data sets. For hourly averages, the coldest temperature measured at PUY is -27.2°C, on 10 February 1956, and the hottest temperature is +30.1°C, on 27 June 2019. This temperature record has been measured when a short heat wave occurred over Western Europe, caused by a mass of hot air coming from the Sahara desert. Average temperatures exceeded by 2°C the regular mean. It has become the hottest June ever recorded in Europe (https://climate.copernicus.eu/record-breaking-temperatures-june). The linear regression line on all points has a slope of +1.4±0.7°C per century (Figure 4). This increase is not far from the increase of almost 2°C of the European air temperature series since the latter half of the 19[th] century, recorded from different datasets of ECMWF; KNMI; NASA; Met Office Hadley Centre, NOAA; and JMA (https://climate.copernicus.eu/surface-temperature).

## 3 Scientific questions

One objective of CO-PDD is to document the evolution of the tropospheric composition over long periods, to quantify the role of anthropogenic emissions in this evolution, and to contribute to better understanding of atmospheric processes driving the observed variability. The localization of the stations and the complementarity of remote sensing and *in-situ* measurements are optimal to evaluate local/regional transport and environmental variability of atmospheric components. The stations are equipped with various sensors and analyzers to document chemical, biophysical, and physical properties of gases, aerosols, clouds and precipitation. The wind tunnel at PUY allows more particularly the study of cloud microphysical properties under natural atmospheric conditions (see Section 5 for technical description). These complex atmospheric processes, especially gas/aerosol/cloud interactions are still poorly understood and parameterized in models (from the detailed process models to the more parameterized chemistry/transport and global models). Those processes have to be studied considering atmospheric transport that is investigated by remote sensing and modeling. More specifically, a panel of key scientific topics exists for all the investigations performed within the CO-PDD activities and that will be continued or developed in the future (Figure 5):

- Long-term evolution of the multiphase chemical composition at CO-PDD allows to analyze the anthropogenic influence, and to characterize/discriminate the temporal evolution of sources (gas/aerosol). In this framework, the location of the PUY station near the interface between the mixing layer and the free troposphere allows to study these time evolutions in different and contrasted situations.

- The configuration of the CO-PDD observation site allows the description of complex interactions between atmospheric components (gas, aerosol particle, water) under natural conditions. Aerosol nucleation processes, gas to liquid partitioning, Cloud condensation nuclei (CCN)/Ice Nuclei (IN) properties of aerosol particles have been for example extensively studied, resulting in instrumental developments.

- The transformation processes of gas-phase precursors to secondary aerosol (SOA) are also actively studied. They can be of a chemical (oxidation processes of the organic matter), physical (microphysical processes linking water vapor, liquid and ice clouds and precipitation) or biological (role of microorganisms in consuming/producing new chemical compounds in the cloud water) nature.

- A better understanding of the precipitation and dynamical context (micro-scale meteorology, vertical atmospheric structure) of these observations is crucial to analyze the atmospheric evolution.

All the above issues are crucial since they are linked to scientific questions related to atmospheric composition, pollution and climate, and their time evolution. CO-PDD has also the objective to provide quality data for model validations and developments (i.e., parameterization for example), as well as for validating satellite remote sensing retrievals.

## 4 French and international structuration

The CO-PDD site has the dual objective to be research platform opened to scientists worldwide and a site operated in the framework of national and international networks. The most significant investments (construction and scientific equipment) have been made through regional strategic development of Region Auvergne (now Auvergne-Rhône-Alpes) and local authorities (department of puy-de-Dôme, Clermont-Ferrand metropole…). Research institutions (CNRS-INSU, CNES, IRSN), and University Clermont-Auvergne also provided substantial support but, as for many research infrastructures, availability of regional funding has been essential.

Beneficiaries from investment funds are the core research institutes operating at the site on a continuous basis: the observatory (OPGC), and the research laboratory (LaMP) for research investigations. The local research framework Environmental Research Federation (FRE) (Clermont Auvergne University) has initiated innovative and cross-disciplinary programs involving the ICCF ("Institute of Chemistry, Clermont-Ferrand") laboratory focusing on interactions between biodiversity and physicochemical processes in the atmosphere.

At national level, the site is recognized as a national observation facility by CNRS-INSU (Institut National des Sciences de l'Univers) in the framework of national infrastructures or services such as ACTRIS-France (Aerosol Cloud and Trace gases Research Infrastructure) grouping activities of AERONET-Photons (AEROsol RObotic NETwork), CLAP (CLimate

Relevant Aerosol Properties), ICOS-France (Integrated Carbo Observation System), OPERA (Permanent observatory of the radioactivity), and RENAG (REseau NAtional GNSS permanent). CO-PDD is identified by the French national space agency (CNES) as a key facility for calibration/validation of space-borne sensors. CO-PDD hosts also internal Observation Services of OPGC, for reactive gases (PUY-GAZ) and for *in-situ* cloud properties (PUY-CLOUD). CO-PDD has also been

involved in a lot of operations connected to research funded under short-term research programs obtained either through the CNRS research program, the National Agency for Research (ANR) or the European Union under FP4, FP5, FP6 and H2020 projects.

CO-PDD is indeed active in the international dimension of the structures described previously in the national context, such as ICOS and ACTRIS (with the ID of PUY). CO-PDD contributes to different networks either research or policy-oriented

and organized at national and international levels, AERONET, EARLINET ("European Aerosol Research LIDAR Network"), EMEP as part of the Convention on long-range Transport of pollutants in Europe, E-Profile of EUMETNET, and GAW (Global Atmosphere Watch). PUY has been labeled as a GAW regional station in 2012 and then GAW global station in 2014. The station is the 30[st] in the world to receive this international recognition and the first in Metropolitan France.

CO-PDD participates to the transnational access program, which houses intensive measurement campaigns carried out by foreign scientific teams. For example, instrumental inter-comparisons of sensors measuring the microphysical properties of liquid clouds have been conducted recently (Guyot et al., 2015), but also campaigns to study aerosols - ice crystals interactions (PICNIC campaign, October 2018) or greenhouse gases spatial distributions (MAGIC campaign, June 2019, see: https://magic.aeris-data.fr/). The data regularly produced by CO-PDD is frequently used for European aerosol

phenomenology studies such as, for example, the study of the chemical (Putaud et al., 2004) and scattering (Pandolfi et al., 2018) properties of atmospheric aerosol particles from ACTRIS sites including PUY or new particle and SOA formation (Dall'Osto et al., 2018). The data of the wind profilers are processed inside the E-Profile network and assimilated as such in the global model of the European Centre for Medium-Range Weather Forecasts (ECMWF).

## 5 Technical description of observation systems

This section presents an overview of *in situ* and remote sensing instruments that characterize the various atmospheric compartments (aerosol particles, gases and clouds). Instruments providing the meteorological context and radionuclides are described in the two last subsections.

### 5.1 Aerosol

### 5.1.1 *In-situ*

After preliminary measurements done during the CIME experiment (Laj et al., 2001), PUY has been increasingly instrumented with *in-situ* aerosol measurements. Specific inlets for aerosol particles sampling also under cloudy conditions

have been deployed. The upper size cut of the whole air inlets (WAI) is at least 30 microns (at a maximum wind speed of 7 m s$^{-1}$), allowing the sampling of a large fraction of cloud droplets (Figure 6a). These cloud droplets are quickly evaporated in the inlet and then sampled by a suite of aerosol instrumentation within the station. Except (NAIS and PSM, all the instruments described here are operated behind a WAI. In contrast, (N)AIS and PSM, which are further described below are dedicated to the monitoring of newly formed aerosol particles with diameters less than 10 nm, and are thus located on the roof of the station where they sample through a shorter inlet (~30 cm, non-heated) to limit diffusion losses.

*Particle size distribution:*

Since 2005, a suite of instruments has progressively equipped the PUY station to monitor particles as well as ion size distributions over a broad size range, from few nanometers up to few tens of micrometers. Particle size distribution in the range between 10 and 400 nm has been measured at PUY since 2005 using a custom-built Scanning Mobility Particle Sizer (SMPS). SMPS inversion is made with the szdist algorithm developed at LaMP and available online (https://hal.archives-ouvertes.fr/hal-01883795). The inversion assumes a theoretical transfer function for the differential mobility analyzer (DMA) and takes into account the Condensation Particle Counter (CPC) efficiency and the charge equilibrium state. It also includes multiple charge correction and accounts for diffusion losses in the instrument. Data quality is regularly checked during inter-calibration procedures and inter-comparison workshops, initially conducted in the frame of the EUSAAR project (European Supersites for Atmospheric Research) and since 2011 within the ACTRIS project (Wiedensohler et al., 2012). The supermicronic fraction of the particle size distribution (0.3 to 20 μm) is measured at PUY with an optical particle counter (OPC, Grimm model 1.108).

Air ion Spectrometer (AIS, Airel Ltd., Mirme et al., 2007) and Neutral cluster and Air ion Spectrometer (NAIS, Airel Ltd., Manninen et al., 2016; Mirme and Mirme, 2013) comprise two identical cylindrical DMA which allow simultaneous measurement of positive and negative ions. Each analyzer operates with high flow rates (sample 30 L min$^{-1}$, sheath 90 L min$^{-1}$) in order to reduce diffusion losses in the sampling lines and further ensure a significant signal to noise ratio, even when ion concentrations are low. In addition to this so called "ion mode", NAIS allows particle sampling in the same manner as the ion measurement, except that in the "particle mode", particles are first unipolarly charged by ions produced by a corona discharge. Each measurement cycle is followed by an offset cycle.

(N)AIS are operated discontinuously at PUY since 2007 to document the lower end of the aerosol size range. They both provide ion size distributions in the range 0.8 – 42 nm in normal temperature and pressure-conditions (NTP), and NAIS allows additional monitoring of particle size distribution between ~2 and 42 nm in NTP-short inlet (length 30 cm), approximately 2 m from the ground, and have been operated on the roof of the chalet since then (11 m above the ground). Good consistency was observed between the data collected with the different AIS and NAIS successively deployed at the site (Rose et al., 2013). Two of our (N)AIS have also been involved in a wider inter-comparison (Gagné et al., 2011), showing that concentrations derived from NAIS measurements could vary from one instrument to the other by up to 10%, which could be assimilated to the uncertainty in the measurement.

In order to complement NAIS observations and get further insight into the first stages of the formation process of neutral particles, measurement of particle concentration in the size range between ~1 and 2.5 nm was initiated in 2012 with a Particle Size Magnifier (PSM, Airmodus Ltd., Vanhanen et al., 2011). The PSM is a mixing type instrument in which the activation of particles results from a rapid and turbulent mixing of the cooled sample flow and heated clean air saturated with diethylene glycol. Counting of the grown particles (mean diameter ~ 90 nm), is then performed with an ordinary CPC (TSI 3010). The activation of a cluster is mainly determined by its size, but its charge and chemical composition affect also the process (Kangasluoma et al., 2013). As a result, there is a systematic uncertainty in atmospheric PSM measurement related to the fact that calibrations of the instrument are performed in the lab with a limited number of known compounds. The instrument was initially operated behind a WAI, and has later been moved on the roof of the chalet (September 2013), close to the (N)AIS, in order to decrease the length of the inlet (individual inlet ~ 30 cm) and further reduce the cluster loss in the sampling lines. PUY is among the first sites equipped with a PSM and measurements from different sites have been recently compared (Kontkanen et al., 2017).

*Optical properties:*

A Multi-Angle Absorption Photometer (MAAP) is in operation at PUY since 2008. MAAP is an optical analyzer combining transmissiometry and reflectometry (Petzold and Schönlinner, 2004). MAAP measures the radiation transmitted and backscattered by the particles impacted on a filter, using a two-flux radiative transfer model to minimize the influence of scattering during absorption measurements. Measurements provided by the instrument are soot carbon (BC) concentrations, from which the absorption coefficient can be traced by multiplying the BC concentration by the specific absorption coefficient (6.6 m² g$^{-1}$).

The TSI 3563 nephelometer measures the scattering of light from a laser source by aerosols and ice crystals. This light is detected at several wavelengths (450, 550 and 700 nm) at different angles by 2 optoelectronic sensors rotating around the sampling volume. This instrument provides integrated measurement of scattered light for angles between 7° and 170° and backscattered between 90° and 170°.The nephelometer has been in operation at PUY since June 2006. The intersection of the particles with the beam produces a diffuse light at the level of the sampling volume, whose analysis makes it possible to obtain the scattering and extinction coefficients, and asymmetry factor of the diffusing elements (aerosols, ice crystals). The combination of the MAAP and the nephelometer makes it possible to calculate the simple scattering albedo $\omega_0$.

*Chemical properties:*

For online measurements an aerosol chemical speciation monitor with a time of flight mass spectrometer (ACSM-ToF) is installed and operating at PUY since 2015. This instrument is capable of measuring the chemical composition of the submicron non-refractory fraction of the organic and inorganic species $NO_3^-$, $SO_4^{2-}$, $NH_4^+$, and $Cl^-$ (Fröhlich et al., 2013). This instrument provides a continuous measurement of aerosol chemical properties with a time resolution from 10 to 30 minutes. The instrument response is determined through calibrations with 300 nm ammonium nitrate particles. These

aerosols are generated from solutions of 0.005 M ammonium nitrate. Aerosols are atomized and then dried to a humidity of < 30% before passing into a differential mobility analyser to select the size (Freney et al., 2019).. (Ng et al., 2011).

Day and night filters are sampled weekly for Organic and Elementary carbon (OC/EC) and major ions measurement. Quartz filters are used to analyze EC/OC with a Sunset analyzer. Cations (sodium, ammonium, potassium, magnesium, and calcium) are analyzed with a Dionex ICS-1500 chromatograph. Concentrations of major water soluble anions (chloride, nitrate, sulfate and oxalate) are determined with a Dionex IC25 chromatograph.

Both the ToF-ACSM and the offline filter (OC/EC) analysis methods are regularly checked through calibration and intercomparison at the European center of aerosol calibration (https://www.actris-ecac.eu/), specifically at the aerosol chemical monitor calibration center for the ToF-ACSM (Freney et al., 2019, Crenn et al 2015)  and at the Joint Research Center (JRC), Ispra for OC/EC, (Cavalli et al., 2010).

*Hygroscopic properties:*

CCN measurements were initiated at PUY in 2011 with a miniature continuous-flow streamwise thermal gradient CCN chamber (CCNc) (Roberts and Nenes, 2005). The ability of aerosol particles to form a cloud droplet is determined using a cloud condensation nuclei chamber (CCNc). Monodisperse aerosol particles (selected using a DMA), are introduced in the CCNc growth column, where they are surrounded by a clean humidified sheath flow. Growth tube walls are continuously wetted with water and the obtained supersaturation is determined by the aerosol flow rate and the column temperature gradient which can both be varied to provide saturations of 0.1 to 0.5%. The CCN number concentration is determined by means of an optical detector at the end of the growth tube, and, for a given size, the activated fraction of the aerosol is inferred from the particle concentration measured by a condensation particle counter (CPC, TSI 3010) operated after the DMA and, in parallel to the CCNc. From 2012 onwards, the DMA and the CCNc have been arranged in a single instrument designed at LaMP called "SCANOTRON", which provides similar measurement as the initial setup but is more compact and eases measurement procedures as well as data analysis. Calibrations of the CCNc are performed on a regular basis using ammonium sulfate and sodium chloride.

The hygroscopicity of aerosols is measured with the Hygroscopic Tandem Differential Mobility Analyzer (HTDMA), an instrument designed at LaMP composed of two DMAs in a tandem arrangement (Villani et al., 2008). This system makes it possible to know the increase in diameter caused by humidity: the hygroscopic growth factor, linked to the chemical composition of the aerosol particles. Ammonium sulfate calibrations and dry scans are performed periodically. HTDMA participated in various international inter-comparisons in the framework of the European EUSAAR project (Duplissy et al., 2009) and showed, during the ammonium sulfate calibration, that the uncertainty of the measured hygroscopic growth factor was 2%.

### 5.1.2 Remote sensing

*Aerosol profiles:*

A Rayleigh-Mie-Raman LIDAR system is in operation at Cézeaux since 2008. This LIDAR is dedicated to the observation of aerosol particles, but also of cirrus clouds and water vapor. It was designed by the Gordien Strato company and built by the Raymetrics company in 2007. The laser source is a Quantel CFR-400 operating in the ultra-violet range (355 nm). The receiving telescope is a 400 mm Cassegrain telescope. The receiving box splits the receiving light in 4 different channels, 2 elastic channels with polarization splitting (parallel and cross), 2 inelastic channels at 387 nm and 408 nm for nitrogen and water vapor Raman scattering. Due to the biaxial configuration of the LIDAR system, the laser beam overlaps completely the field of view of the telescope at about 1000 m above the LIDAR, and is in a partial overlap at about 500 m. Since 2013, the operation is automatized to operate continuously the elastic channels during day and night, and the inelastic channels during night only for appropriate conditions. The LIDAR produces profiles of backscatter signal, from which level 2 data are calculated: water vapor mixing ratio, and aerosol/cirrus extinction and backscattering coefficient (Fréville et al., 2015). In 2018, the COPLid improvement project (Figure 6d) made it possible to change the laser source for a more powerful laser, to optimize the infrastructure of the LIDAR inside the building of the University, and to add acquisition channels at 532 and 1064 nm wavelengths for aerosol-cirrus measurements. In this up-to-date configuration, the Cézeaux LIDAR system reaches the optimal configuration requested by EARLINET standards, allows to retrieve aerosol microphysics and is more appropriate to compare aerosol and cloud products with present and future satellite retrievals as CALIOP (Cloud-Aerosol LIDAR with Orthogonal Polarization, since 2006) onboard CALIPSO and ATLID (Atmospheric LIDAR) onboard EarthCare (Earth Clouds, Aerosols and Radiation Explorer, launch planed in 2021).

*Aerosol vertical columns:*

A CIMEL CE318 automatic sun tracking photometer is in operation at Cézeaux since 1999 in collaboration with the LOA (Laboratoire d'Optique Atmospherique, Lille, France) and in the framework of the AERONET international network (Holben et al., 1998). The photometer measures the solar radiance in 5 spectral bands: 440, 670, 870, 936, and 1020 nm. The spectral band of 870 nm is measured from 3 polarized channels of 120 degrees. This instrument provides the integrated optical thickness of atmospheric aerosols, and the Angström coefficients calculated from the direct solar radiance. It can also retrieve the volume size distribution (fine and coarse modes), through multi-angular solar radiance measurements with an inversion algorithm detailed in (Dubovik and King, 2000).

## 5.2 Greenhouse and reactive trace gases

### 5.2.1 *In situ*

*Greenhouse gases*

Greenhouse gases ($CO_2$, $CH_4$, $N_2O$ and $SF_6$) measurements in November 2000 as part of the French monitoring program SNO-RAMCES/ICOS-France (Broquet et al., 2013; Lopez et al., 2015; Ramonet et al., 2010; Sturm et al., 2005). Three

types of analyzers have been used: non-dispersive infrared instruments (NDIR, LICOR-6251 from 7 November 2000 to 20 November 2007; LICOR-6252 from 21 November 2007 to 03 April 2011); gas chromatography (GC, Agilent HP-6890N from July 2010 to April 2015); and cavity ring-down spectrometers (CRDS) commercialized by PICARRO (G1301/#76 from 3 April 2012 to 2 February 2015; G2401/#285 from 2 February 2015 to 25 August 2016; G2401/#473 since 25 August 2016).

The automated gas chromatograph (GC) system (Agilent HP-6890N) was equipped with a flame ionization detector (FID) with a nickel catalyst for $CH_4$ and $CO_2$ detection and a micro electron capture detector (µECD) for $N_2O$ and $SF_6$ detection (Lopez et al., 2015). Two working standards, each calibrated against the WMO scale, were injected every 30 min to correct for instrumental drifts. Data quality control was insured through hourly injections of a target gas and regular instrumental calibrations were performed using three WMO calibrated compressed air cylinders. Repeatability is 0.1 ppm for $CO_2$, 1.2

340 ppbv for $CH_4$, 0.3 ppbv for $N_2O$ and 0.06 pptv for $SF_6$.

Following the ICOS standard protocol, the Picarro instruments are calibrated upon a suite of 4 calibrated compressed air cylinders provided by the ICOS Calibration center (calibrated against the WMO scale) every two to four weeks, and quality control of the data are ensured by regular analysis of two target gases (with known and calibrated concentrations); one short term target gas analyzed for 30 min at least twice a day and one long term target gas analyzed for 30 min during the

345 calibration procedure (enabling a long term -10 years at least- data quality control).

For all instruments, ambient air is pumped from the roof platform through a 10 m long dekabon tubing and dried via a glass trap cooled by a cryocooler at -60°C prior injection into the analyzer. The drying system has been suppressed in August 2016 with the installation of the CRDS, allowing measurement of atmospheric moisture content. The Picarro analyzer enables a measurement of atmospheric moisture content, which is used to correct the measured GHG concentrations.

The GC setup allows analyzing approximately five atmospheric samples per hour, whereas the infrared instruments (NDIR, CRDS) allow continuous measurements (minute or hourly averages are calculated from raw data measured every 2.5 sec).

In addition to the continuous measurements, glass flasks are weekly sampled for $\delta O_2/N_2$ and $CO_2$ analysis at the Physics Institute, University of Bern (Sturm et al., 2005; Valentino et al., 2008), and for $CO_2$, $CH_4$, $CO$, $N_2O$, $SF_6$, $H_2$ measurements at the laboratory LSCE. Isotopic content measurements of $CO_2$ ($\delta^{13}C$ and $\delta^{18}O$) have also been performed on the flasks

samples at LSCE until 2016. $CO_2$ and $CH_4$ measurements at PUY have been certified by a GAW audit in April 2016. During this audit, the station also hosted the ICOS mobile laboratory which is circulating within the ICOS network in order to evaluate the compliance to the ICOS protocols.

*Ozone measurement:*

Since 1995 a UV photometric analyzer (from 1995 to 2003: a model commercialized by Environment SA; from 2003 to

360 2017, a Thermo Scientific model TEI 49c; since 2017, a Thermo scientific model TEI 49i) has provided ambient

measurements of concentrations of ozone ($O_3$). The Model 49i uses a dual-cell photometer, the concept adopted by the National Institute of Standards and Technology as the principle technology for the national ozone standard. It measures the amount of ozone from 0.05 ppb concentrations up to 200 ppm (response time of 20 s, precision of 1 ppb). The calibration is performed every 3 months using a dynamic standard mixture generator (ANSYCO), which is certified and traceable to the national calibration chain for air quality. Therefore, it has been checked in the framework of a GAW audit in 2014.

*$NO_x$ and $SO_2$ measurement:*

The measurement of $NO_x$ ($NO + NO_2$) has been deployed at PUY since 2003 using an analyzer by chemiluminescence (from 2003 to 2018, a Thermo scientific model TEI 42cTL; since 2019, a Thermo Scientific model 42iTL). The instrument measures directly NO and indirectly $NO_x$ after conversion into NO. Until 2012, the converter was a molybdenum one which is not selective for $NO_2$ since its converts also part of other $NO_y$ (especially HONO, $HNO_3$, and PAN). Since December 2012, a Blue Light Converter (BLC) specific for the selective conversion of $NO_2$ equips the analyzer. Tendencies have been analysed separately for $NO_2$ during the period 2003-2012, ($NO_2$ + part of $NO_y$ converted) and 2012-present, which is real $NO_2$. Since, the $NO_x$ measurement protocol follows the ACTRIS standard operation procedure. Datasets are validated each year and are available on the EBAS database website (http://ebas.nilu.no). $SO_2$ concentrations are measured since 2003 with a Thermo Scientific model 43i, changed for a Thermo Scientific model 43iTL in 2017. NO, $NO_2$ and $SO_2$ are calibrated every month with reference mixtures    obtained from the dynamic dilution of working standards (at 5 ppm), which are checked every years with traceable standards. The $NO_x$ instrument took part in a side by side intercomparison held in Hohenpeissenberg in 2012 in the frame of the ACTRIS program.

*VOC measurement:*

Active sampling on sorbent cartridges is performed at PUY since 2010 to measure Volatile Organic Compounds (VOC). For this, a smart automatic sampling system (SASS) developed by TERA Environment is used. Gaseous compounds are sampled at approximately 6 m above ground level, using a Teflon sampling line, and then trapped into a Tenax® sorbent cartridge at a flow rate of 100 ml min$^{-1}$. The time duration of sampling varies between 40 min to 2-3 hours and allows the trapping of major anthropogenic and biogenic VOCs: $C_5$-$C_{14}$ n-alkanes, alkenes, terpenoids and aromatic compounds. The analytical device used for the cartridge analysis consists in a gas chromatograph - mass spectrometer system (GC-MS, Perkin Elmer) connected to an automatic thermal desorber (Turbomatrix Perkin Elmer). Storage times for the cartridges is less than 3 months before analysis by GC-MS. Cartridges have Swagelok caps, and are kept in a dark and cool room (~20 °C) at the laboratory. The analytical conditions are described in other recent articles (Dominutti et al., 2019; Wang et al., 2020). Since September 2018, an on-line GC-FID (gas chromatography - flame ionization detector) has been running at PUY for the on-line monitoring of non-methane hydrocarbons (NMHC) from C2 to C10. Air samples collected every 2 hours at 25 mL min$^{-1}$ during 40 minutes are dried through a semipermeable membrane (Nafion, Permapure Inc.) and introduced in a trap cooled by Peltier effect. This system is an adaptation of the Perkin Elmer Turbomatrix coupled with the GC-FID 6890 from Agilent.

VOCs measurements follow the ACTRIS/GAW Standard Operating Procedures for sampling, calibrations and data validation. PUY has participated to an inter-comparison exercise through a round-robin test (Hoerger et al., 2015).

### 5.2.2 In-situ measurements along the PUY slopes

In 2012, the tourist train line called "Panoramique des Dômes" was opened and joined the base of the puy de Dôme to its summit. It has been equipped with a sampling platform containing an ozone analyzer (2B technology), temperature, pressure
and humidity sensors (Figure 6c). This information is used to follow the concentration gradient along the slope of the puy de Dôme and to determine the mixing layer height and the inversion temperature that occurs along the slope of the volcano.

### 5.3 Cloud

### 5.3.1 *In-situ*

*Microphysical measurement:*
Gerber PVM-100 is a ground-based- scattering laser spectrophotometer for cloud droplets volume measurements and is manufactured by Gerber Scientific, Inc., Reston, Virginia. The laser light emitted at the wavelength $\lambda = 0.780$ µm is dispersed by cloud droplets passing through the sampling volume of the 3 $cm^{-3}$ probe. This light is collected by a lens system whose angle varies from 0.25 to 5.2°. The scattered light is converted into a signal which is proportional to the droplets density (or LWC) and the particle surface density (PSA) with a time resolution of 5 minutes (Gerber, 1984, 1991). It has
been tested and inter-compared with other instruments during ACTRIS transnational access activities at PUY in May 2013 (Guyot et al., 2015).

*Cloud sampling:*
Since 2001, cloud water sampling is regularly performed at PUY using a dynamic one-stage cloud water impactor (Brantner
et al., 1994), with a cut off diameter of approximately 7 µm. Recently, a new cloud water collector was developed to increase the efficiency of collection (Figure 6). Before sampling, the aluminum impactor is cleaned and sterilized by autoclave to allow micro-biological investigations. A fraction of the cloud water sample is filtered immediately after collection to eliminate microorganisms or particles, while another fraction is kept unfiltered for microbiological measurements. Different bio-physico-chemical parameters are determined systematically on cloud water samples filtered and non-filtered fractions.
Some bio-chemical parameters are immediately measured on site from the fresh samples, while others can be performed later in the laboratory from stabilized and/or frozen samples. The measurements are used to evaluate the variability of the cloud water composition that depends on both season and air mass origin.

*Cloud chemical measurements:*

The pH is determined immediately after sampling. Main inorganic cations ($Na^+$, $NH_4^+$, $K^+$, $Mg^{2+}$, $Ca^{2+}$) and anions ($Cl^-$, $NO_3^-$, $SO_4^{2-}$, $PO_4^{3-}$) as well as short chain carboxylic acids (formate, acetate, succinate, oxalate) are measured by ion chromatography (IC). An analyzer measuring the total organic carbon concentration is used to estimate the organic matter present in cloud water. Hydrogen peroxide ($H_2O_2$), a strong oxidant in the cloud water, is determined by spectrofluorimetric quantification method (Li et al., 2007). The concentration of Fe(II) and Fe(III), a key parameter for the cloud water oxidative

capacity evaluation (Deguillaume et al., 2005), is determined by a spectrophotometric method after chemical complexation (Stookey, 1970).

More recently, the analysis of the chemical composition of the cloud waters sampled at the PUY has been improved with the quantification of 16 amino acids (AAs) determined using a new complexation method coupled with high-performance liquid chromatography (HPLC) (Bianco et al., 2016); the concentrations of 33 metal elements have been determined using

Inductively Coupled Plasma Mass Spectrometry (ICP-MS) (Bianco et al., 2017). The oxidative capacity of the cloud water has also been evaluated following the hydroxyl radical (HO$^\bullet$) formation rates during the irradiation of cloud waters under sun-simulated radiation (Bianco et al., 2015).

Finally, ultrahigh-resolution mass spectrometry has been recently used to get a better identification of the dissolved organic compounds. Using GCxGC-HRMS technique, more than 100 semi-volatile compounds were detected and identified

(Lebedev et al., 2018). Among them, phenols and phthalates that are strong pollutants were quantified. Ultrahigh-resolution Fourier-transform ion cyclotron resonance mass spectrometry (FT-ICR MS) has also been used to identify a wide spectrum of organic compounds (up to 5000 assigned molecular formula) that have been shared into several classes depending on their H/C and O/C ratio (Bianco et al., 2018, 2019a).

*Cloud biological measurements:*

Since 2003, microorganisms in cloud waters and their activity are also investigated (Amato et al., 2005; Vaïtilingom et al., 2012). Each cloud water sample is collected under sterile conditions and microbiological parameters such as biomass, biodiversity and biological activity are measured. Total cells are counted using epifluorescence microscopy (before 2010) or by flow cytometry. Bioluminescence is used to quantify adenosine triphosphate (ATP) as a marker of the whole metabolic

activity. Living cultivable microorganisms (bacteria and fungi) are generally investigated as well, by platting samples on nutritive media and then isolated and identified against databases by 16S rRNA gene sequencing (Amato et al., 2007b; Vaïtilingom et al., 2012). This allows the isolation of microorganisms from the cloud water, and their interaction with atmospheric chemical compounds (Amato et al., 2007a). Now more than 1000 microbial strains are available for in-lab experiments and for biotechnological applications. Recently, molecular biological techniques based on DNA and RNA have

been developed to study microbial diversity and activity in clouds (Amato et al., 2017, 2019).

### 5.3.2 Wind tunnel

Located at PUY, the wind tunnel is a unique national facility that operates under natural atmospheric conditions (cloudy conditions or not). It was built and installed in 2010 during the renovation of the station. The wind tunnel (Figure 6e) has an open circuit configuration with an entrance located on the west side of the chalet facing prevailing winds. The three-phase electric motor (132 kW) generates a variable air flow up to 17 m$^3$ s$^{-1}$ (variable and controlled speed). Depending on the type of application, two interchangeable rectangular section tubes allow different maximum air velocity, depending on the objective: the tube with a section of 250 × 320 mm and a length of 600 mm allows a maximum speed 120 m s$^{-1}$, the section of 540 × 640 mm and of length of 2000 mm allows a maximum speed 52 m s$^{-1}$.

The wind tunnel offers several applications:

- for scientific issues: development of original methodologies and equipment for cloud and aerosol *in-situ* aircraft observations for studies on cloud / chemistry / climate interactions,

- for technological issues: test and validation of airborne instruments; the wind tunnel has been used for example in May 2013 for an evaluation campaign of optical sensors for cloud microphysical measurements (Guyot et al., 2015). The wind tunnel is integrated in validation procedures of airborne microphysics probes,

- for industrial applications: the wind tunnel has been used for studies of aircraft elements in icing conditions in the context of flight certifications, or at the requested of the aviation industry.

### 5.4 Precipitation, meteorology, dynamics

CO-PDD is a sustainable site for the observation of precipitation at the scale of the urban basin of Clermont-Ferrand since 2006. For this purpose, a combination of RADARs are operating at different frequencies in the K and X band, providing the size spectra of hydrometeors and spatial distribution of reflectivity, as well as optical disdrometers and a network of rain gauges. The dynamical and meteorological context is characterized using *in-situ* and remote sensing measurement of wind, temperature and humidity.

### 5.4.1 *In-situ*

*Meteorological parameters:*

The meteorological context of CO-PDD is completed by ground level meteorological stations and sonic anemometers operating continuously at all the measurement sites of CO-PDD. Near ground humidity measurements are available at Cézeaux since 2002 and at PUY since 1995. The meteorological sensors of the Cézeaux and puy de Dôme sites are Vaisala HMP45 (HUMICAP 180 humidity sensor and PT100 temperature sensor). The accuracy and other technical specifications are available on the Vaisala manufacturer's website (https://www.vaisala.com/sites/default/files/documents/HMP45AD-User-Guide-U274EN.pdf).

*Raindrops size distribution:*

Parsivel² is an optical disdrometer commercialized by the OTT Company and designed to individually measure the diameter and fall speed of the rain drops. This measurement is made when drops intersect a laser beam having a final sampling surface of 54 cm². The diameter of droplets is estimated from the decrease of the intensity of the laser beam received by a photoelectric diode, and the fall speed is estimated by the time taken by the drop to cross the beam. Two Parsivel² instruments are deployed at Aulnat and Opme, in complement with a network of classical rain gauges and in conjunction with MRR's in order to provide continuous DSD profiles from the ground up.

*Full Sky imager and webcam cameras:*

Two CO-PDD webcams are in operation, one at Cézeaux looking towards PUY and the other at PUY looking towards Clermont-Ferrand area. These are Axis P1343 high performance cameras, and widely used in daytime and nighttime video surveillance. They capture a 600 × 800 pixels image every ten minutes in the compressed JPEG format. More technical details on the cameras are available on the Axis manufacturer's website (https://www.axis.com/en). In addition, an full sky imager is in operation at Cézeaux since December 2015. This is an EKO SRF-02 camera, a fully automatic imaging system that captures 2 mega-pixels JPEG images of the total sky every two minutes at two different time exposures. When sky images are captured, cloud fraction is automatically calculated by the software, which gives much flexibility and functions to define the area of interest by horizon masking. The SRF-02 is connected to a computer network through the standard Ethernet interface. The full Sky imager is equipped with an air pump and a drying cartridge. All sky images are processed with the ELIFAN algorithm which aims at estimating the cloud cover amount (Lothon et al., 2019).

### 5.4.2 Remote sensing

*Rain spatial distribution:*

A X band local area weather RADAR is operated in order to provide 2-D reflectivity maps. The original system is a modified navigation RADAR designed by the University of Hamburg and the Max Planck Institute for Meteorology and has been operated in Clermont-Ferrand since fall of 2006. The frequency of this system is 9410 MHz and the current emitted peak power 24 kW. The antenna is an off-set parabola with a diameter of 90 cm to define a suitable pencil beam for precipitation observation. Records are accumulated over successive rotations of the antenna which has a scan rate of 24 revolutions per minute. Another X band RADAR system commercialized by the ELDES Company (WR-10 X) has been purchased in 2011 and installed on board a mobile trailer. This RADAR operates at the same frequency, with an emitted peak power of 9.5 kW, and a 70 cm diameter center feed antenna. As opposed to the previous system which operates plan position indicator (PPI) scan at a fixed pre-set elevation, the ELDES RADAR can perform multiple elevation volume scans as well as range height indicator (RHI) scans. The typical operation mode for the two RADAR systems corresponds to below 1 minute time integration and below 100 m range resolution with a maximum range between 20 and 36 km according to experimental and weather conditions. These RADAR systems have been involved in the COPS (Convective Orographically-

520 driven Precipitation Study), (Hagen et al., 2011; Van Baelen et al., 2011; Wulfmeyer et al., 2011) and HYMEX (HYdrological cycle in Mediterranean Experiment) (Zwiebel et al., 2016) international campaigns. In 2019, due to technical problems, the older RADAR has been removed from the Cézeaux platform and replaced with the ELDES system.

The two MRRs (Micro Rain RADAR) in operation at Opme and Aulnat are K-band (24.1 GHz) vertical Doppler FMCW RADARs commercialized by the METEK Company. They allow the investigation of the vertical microphysical structure of
525 precipitation. Measurements from the retrieved reflectivity and Doppler velocity profiles make it possible to reconstruct the profiles of rain rate and drop size distribution through drop sorting with the Atlas relationship between drop size and fall speed (Atlas et al., 1973), over 32 range gates with 100 m vertical resolution up to 3000 m every 10 s. The spectral reflectivity is dynamically analyzed to estimate and remove the noise, and the attenuation is corrected along the path by using an improvement of the Hitschfeld and Bordan algorithm (Tridon et al., 2011). The peak power is 50 mW, the sampling
frequency is 125 kHz and the antenna diameter is 60 cm.

*Wind profiles:*

Wind profiling RADARs use the Doppler frequency shift of signals scattered from atmospheric turbulence to monitor wind profiles. One system is operating at Opme in the VHF band (45 MHz), since 1999. The antenna field is made of two
perpendicular sets of 60 m coaxial–collinear lines fed with or without fixed delay, enabling the RADAR beam to be successively pointed vertically, and in four oblique angles, depending on a control sequence. This allows the measurement of the time evolution of the horizontal and vertical wind profiles from 2 to 12 km a.s.l. with a vertical resolution of 375 m (Baray et al., 2017).

To complete the wind profile monitoring of the lower troposphere, a refurbished RADIAN prototype system has been in
operation at Aulnat, near Cézeaux, since 2014. This profiler is based on the same principle that the VHF wind profiler, but it is operated in the UHF band. The micropatch antenna of 2.8 × 2.8 m is controlled sequentially in 5 directions. That allows the measurement of the time evolution of the vertical and horizontal wind from 500 to 3300 m a.s.l. with a vertical resolution of 100 m.

*H₂O profiles and columns:*

As mentioned in Section 5.1.2, the LIDAR system in operation at Cézeaux is equipped with Raman channels for nitrogen and $H_2O$, providing vertical profiles of water vapor from 1 to 10 km of altitude.

Since the 1990's, the GPS (Global Positioning System) has proven to be an autonomous, all-weather and continuous system for the measurement of atmospheric water vapor (Bevis et al., 1992). The GPS, GLONASS (GLObal NAvigation Satellite
System) and GALILEO satellite constellations signals are collected by ground based receivers. With respect to propagation in a vacuum, the signal traveling between a GPS satellite (altitude of 20200 km) and a ground-based receiver is delayed by the atmospheric constituents (dry air, and water vapor). The zenithal wet delay, due mainly to water vapor abundance, can be estimated from the difference between the total atmospheric zenithal delay and its hydrostatic term, i.e., the zenithal

hydrostatic delay, or so-called dry delay, which depends on the total weight of the atmosphere above. The zenithal wet delay is converted into the integrated water vapor column with a good accuracy, using surface temperature and empirical formulas (Bevis et al., 1992). GPS receivers are in operation at Cézeaux, PUY and Opme. Set at different altitude levels, they allow the monitoring of the water vapor evolution in the different corresponding atmospheric layers sampled.

## 5.5 Radionuclides

Man-made and cosmogenic radionuclides attached to aerosols are studied as part of the French Observatoire Permanent de la Radioactivité (OPERA program) of the "Institut de Radioprotection et de Sûreté Nucléaire" (IRSN). Anthropogenic and naturally occurring radionuclides are monitored on a monthly basis in rainwater sampled at Opme. Aerosols are also sampled on a weekly basis at PUY since 2005 and Opme since 2006. PUY is one of the two highest radionuclide monitoring stations in France.

Detection of radionuclides at trace levels requires high sampling volume (several tens of liters) or collection surface (1 to 3 $m^2$). The summit is equipped with a high volume sampler (max sampling rate of 700 $m^3$ $h^{-1}$) since 2010 and using an electret polypropylene fiber filter to ensure high collection efficiency (minimum of 95% for 30 nm diameter). Aerosol filters are analyzed by gamma spectrometry on low-background high-purity Germanium (HPGe) detectors in an underground laboratory. Another cloud water collector is deployed at PUY to collect radionuclides. A Caltech type cloud droplet sampler is used to collect high volume in order to exceed the detection limits. Up to 80 liters can be sampled on a monthly basis. The sampler is equipped with heating rods to allow sampling in winter conditions using sampling-heating cycles. Radionuclide characterization at the PUY strengthens the European monitoring of airborne radioactive contamination at high altitude as for instance at the Sonnblick, Jungfraujoch or Zugspitze stations which play an important role in the knowledge of trans-boundary dispersion of radioactive plumes, such as after the Fukushima accident or during European-scale events of unexpected radionuclides in the atmosphere (Masson et al., 2016).

## 6 Main scientific results

### 6.1 Trace gases

An increase of anthropogenic greenhouse gases (GHG) concentrations is observed in the atmosphere, leading to a modification of their natural cycles and to a strong increase in atmospheric radiative forcing. Accordingly, the concentrations of $CO_2$ have regularly increased since the start of measurements at PUY (Figure 7). The average $CO_2$ mole fraction has increased from 370 ppm in November 2010 up to 410 ppm in November 2018 (+11%). This is consistent with observations at other locations (example: Mauna Loa 409.5 ppm, obtained at an altitude of 3400 m in the remote northern subtropics, November 2018). At the same time, the mean mole fraction of $CH_4$ has also increased from 1.850 ppm to 1.950 ppm during the same period (not shown). A 3 years (July 2010 - July 2013) analysis of GHG measurements has been performed using a gas chromatograph system located at PUY (Lopez et al., 2015). The analysis of the 3-years atmospheric time series revealed

how the planetary boundary layer height drives the concentrations observed at PUY. Radionuclide measurements are used to determine the boundary layer / free tropospheric conditions (Farah et al., 2018). The $CO_2$ surface flux are estimated and revealed a clear seasonal cycle, under the influence of plant assimilation, and burning of fossil fuel (Lopez et al., 2015, Ramonet et al., 2020). According to Lopez et al., 2015, the measurements observed at PUY during the night are representative of the central part of France, mostly west of the station. Similarly to other European mountain sites like Schauinsland or Monte Cimone, the daytime values are more influenced by local sources, and therefore they are generally excluded in the large scale atmospheric inversions (Broquet et al., 2013; Bergamaschi et al., 2017).

Reactive gases concentrations such as ozone have, however, decreased slightly but not significantly over the past 20 years (-0.1±1.2 ppbv/decade, Figure 8). Expected marked seasonal variations (more $O_3$ in summer due to photochemistry, less in winter) are in line with observations in Europe since 1990 (Gilge et al., 2010; Jonson et al., 2006), while its primary precursors like $NO_x$ has significantly decreased (not shown). This slight decrease of ozone since 20 years is consistent with the ozone trend reported at surface stations within the EMEP network between 1990 and 2012. Changes in long-range transport, a reduced titration by NO due to less $NO_x$ availability and higher biogenic emissions in a warming climate could explain these trends (Colette et al., 2016). Due to their complexity, the relationships between $O_3$ precursors need further supplementary analyses that are currently underway.

On the contrary, sulfure dioxide has dropped significantly in the last 15 years (-0.23±0.05 ppbv decade$^{-1}$) due to the reduction of its primary emission. This is consistent with SO2 trends observed over Europe (Hohenpeissenberg, Giannitrapani et al.,2006). The monitoring of NMHC is too recent for deriving multi-year trends but shows the systematic presence of major anthropogenic and biogenic NMHC with levels in the same range as the ones reported at European GAW stations like Monte Cimone and Hohenpeißenberg stations (Wang et al., 2020). Due to the diurnal and seasonal cycles of the boundary layer height, compounds influenced by anthropogenic sources are more concentrated in summer than in winter when PUY is in free troposphere. Local and regional emissions will influence the trace gas concentrations especially in summer and daytime.

## 6.2 Aerosol

Studies of microphysical, chemical and dynamical atmospheric processes are carried out using CO-PDD measurements. They cover the whole life cycle of aerosol particles from their formation by nucleation from gaseous precursors, chemical processing during transport, activation to cloud droplets to washout from precipitation.

The total particle number concentration (> 10 nm) currently measured at PUY is on average ~ $2\times10^3$ cm$^{-3}$, which corresponds to intermediate values compared to observations reported from neighboring mountain stations in Europe (Laj et al., 2020), such as for instance Montseny (Spain, 700 m a.s.l., ~ $3\times10^3$ cm$^{-3}$) or Jungfraujoch (Switzerland, 3578 m a.s.l, ~ $2\times102$ cm$^{-3}$). As illustrated on Figure 9, the aerosol number concentration tends to overall exhibit a slight decrease over the past 15 years at PUY, in the order of -9 ± 5 $\times10^2$ cm$^{-3}$/decade. Deeper investigation of this trend is currently performed and will include a more detailed discussion of these aspects. The variability of atmospheric aerosol number concentration at PUY

show a marked seasonal variation with a maximum during the summer, consistent with observations from other high altitude sites where a stronger seasonal contrast is usually observed compared to continental lowland sites (Laj et al., 2020). A daily cycle is also very clear, with peak concentrations during the day. These seasonal and daily variations can be explained by a combination of factors including a fundamental role of the dynamics of the planetary boundary layer and its exchanges with the free troposphere, as well as a non-negligible impact of nucleation (or new particle formation (NPF) events) on the particle size distribution (Venzac et al., 2009). In fact, the dynamics of the planetary boundary layer, in connection with the altitude and the topography of the sites, plays a major role in the transport of both pre-existing particles and gaseous precursors at high altitude, and in turn significantly contributes to the differences observed among mountain stations (Collaud Coen et al., 2018).During the day, the boundary layer (BL) height increases until reaching PUY, transporting at altitude the aerosols emitted from the surface. This vertical transport by convective mixing is more marked in summer than in winter. At night, the concentrations are more representative of the free troposphere (FT) / residual night layer (RL), and the database acquired for several years provides particle size spectra of background air masses. It is clear that the nocturnal residual layer is largely influenced by the diurnal boundary layer as evidenced by the seasonal variability of the concentrations persisting at night. A more detailed segregation of BL/FT air masses using a set of measurements and models outputs shows that the size distribution actually keeps the air mass type signature even in air masses transported in the FT for more than 75 hours (Farah et al., 2018).

The frequency of new particle formation events is on average 30% of the measurement days (Rose et al., 2013; Venzac et al., 2007), which is a relatively high frequency compared to most European low-troposphere environments (Manninen et al., 2010). A key question is to characterize the vertical extension of these new particle formation events. Size distribution measurements performed simultaneously at PUY and OPME show that for over 45% of the time, NPF events are occurring at high altitude while not occurring at low altitude. Such situation is mostly observed when the planetary boundary layer height derived from LIDAR measurements performed at Cézeaux indicates that the PUY station is close or within the lower FT. The remaining observation show that NPF occurs over the entire atmospheric boundary layer (Boulon et al., 2011). This enhanced NPF frequency at high altitude is observed despite the fact that clouds, often found at PUY, are inhibiting the NPF processes as they represent a large condensation surface that acts as a sink for aerosol embryos and condensable gases (Venzac et al., 2007).

Specific measurements in the free troposphere show that the process of forming neutral clusters dominates over the formation of ionized particles (Rose et al., 2015), but overall ion induced nucleation is promoted at high altitudes compared to low altitudes (Sellegri et al., 2019).

Observations of the aerosol chemical fractions further improved our knowledge of the sources and processes of aerosol transformation in the atmosphere. We find that anthropogenic emissions have a limited impact on PUY measurements, making PUY representative of a rural background site. The organic compounds represent a large fraction of the aerosol mass at aerosol diameters less than 400 nm observed at PUY (Freney et al., 2011; Sellegri et al., 2003b). The seasonality of aerosol mass is similar to the one found for aerosol number, showing highest mass concentrations measured during the

spring and summer months and lowest concentrations during the winter months (Bourcier et al., 2012b). Aerosol chemical composition monitored at PUY is highly variable but average concentration monitored at PUY over the period April 2015 – February 2016 exhibits the following values: organic 57% (2 µg m$^{-3}$), followed by sulphate 16% (0.4 µg m$^{-3}$), nitrate 12%

(0.3 µg m$^{-3}$), ammonium 10% (0.24 µg m$^{-3}$) and BC 5% (0.13 µg m$^{-3}$) (Farah et al., 2020). In particular, the organic fraction is higher in summer because of additional sources of secondary organic carbon at altitude with respect to the boundary layer in summer. The organic aerosol detected at PUY can be broken down into three fractions of different origins: a dominant fraction of oxygenated semi-volatile compounds representative of aged aerosols and transported over long distances, a fraction of organic compounds from biomass combustion, significant in winter, and a fraction of primary organic which is

minor (Freney et al., 2011, Farah et al., 2020).

One of the impacts of aerosol particles on the climate is that it scatters and absorbs solar and telluric radiation. These interactions depend on the concentration, size and chemical composition of the atmospheric aerosol, but they can be directly measured in terms of scattering and absorption coefficients. A median scattering coefficient of ~ 10 Mm$^{-1}$, in the range of values observed at other mountain sites, was obtained by Pandolfi et al. (2018) for the period 2007-2014 at PUY. Seasonal

medians in the range 0.7 – 9 Mm$^{-1}$ were in addition more recently reported by Laj et al. (2020) for the year 2017, together with median absorption coefficients of 0.92 and 0.44 Mm$^{-1}$ for spring and autumn, respectively. The climatology of these scattering and absorbing properties shows that, quite logically, compared to the results presented previously, the height of the boundary layer strongly influences the optical properties of the aerosol at puy de Dôme, since it influences the concentrations in number, mass concentrations and chemical composition. This influence is reflected in clear daily and

seasonal variations in optical properties, when considering the whole long term data set of optical properties. A significant decrease of the daytime aerosol scattering coefficient during summer and winter was in particular observed by Pandolfi et al. (2018) over the period 2007-2014. The recent results from Collaud Coen et al. (2020) are however slightly balancing the previous findings of Pandolfi and co-workers, as an overall decrease of -0.147%/year over the decade 2009-2018 was observed but reported as not statistically significant. A statistically significant decreasing trend was in contrast found for the

absorption coefficient over the period 2009-2017, in the order of -0.017%/year. These observations areare in agreement with aerosol mass (PM) decay observed at European level, in relation to SO$_2$ emission regulations. *In-situ* optical data were also used to evaluate aerosol remote sensing retrievals. *In-situ* measurements of aerosol optical and size properties measured at PUY can be combined with integrated sun-photometer measurements and LIDAR profiles to assess the accuracy of inversion algorithms for retrieving aerosol size distributions, showing a fairly good agreement between *in-situ* and column average

aerosol size distribution when the structure of the atmosphere is taken into account (Chauvigné et al., 2016).

In order to evaluate both the climate and health impacts of particles in the atmosphere, it is essential to know the quantity of water that the aerosol particles contain at a given relative humidity level (i.e,. hygroscopicity). Compared to measurements made at other elevation sites, the hygroscopic properties of the aerosol sampled at PUY are intermediate between that of the aerosol sampled at higher altitudes in the Alps (Jungfraujoch, 3580 m a.s.l., Switzerland) (Sjogren et al., 2008), and aerosols

sampled at a lower altitude at Great Dun Fell (848 m a.s.l., England) (Svenningsson et al., 1997), or Kleiner Feldberg (878 m

a.s.l., Germany) (Svenningsson et al., 1994). Lower-elevation sites contain a larger fraction of hydrophobic aerosols from the lower layers of the atmosphere, and higher elevation sites or ocean-influenced air masses contain a larger fraction of highly hygroscopic aerosols (sea salts). This is reflected by the observations performed at PUY, where ~ 45% of the sampled air masses originate from oceanic regions and tend to contain more hygroscopic particles compared to other sectors (Holmgren

et al., 2014).

Regardless of the air mass type, the moderately hygroscopic particles (hygroscopic growth factor close to 1.4) dominates the entire aerosol number concentration, illustrating the effect of ageing of the aerosol during transport.

Lastly, CCN measurements performed at PUY show that CCN concentrations are highest in continental air masses compared to oceanic air masses, because these air masses contain higher particles number concentrations, and also a higher proportion

of inorganic (hence hygroscopic) compounds. The aerosol activation diameter at PUY is generally close to 100 nm at the super-saturation of 0.24%, and the number concentration of particles larger than 100 nm is a fairly good approximation of the number of cloud droplets (Asmi et al., 2012). The mixing of the aerosol is however influencing the way each chemical component is entering cloud droplets. It has been shown that the activated fraction of inorganic aerosol particles is 0.76 for 200 nm particles, and 0.93 for 500 nm particles, while it is only 0.14 for organic species at all sizes of particles (Sellegri et

al., 2003a). It was also found that elemental carbon (EC) has a higher activation fraction than organic carbon. These results suggest that inorganic and organic species are externally mixed, while EC has likely experienced internal mixing with inorganic species during the course of transport to the site.

## 6.3 Cloud

In a similar way as for the atmospheric aerosol, cloud studies, from the mechanisms driving the activation of cloud droplets

and the cloud microphysical properties, to the in-cloud chemical and biological processes and their impact on atmospheric chemistry have been among the main topic studied at puy de Dôme since several decades. Clouds are frequently observed at the PUY station (Baray et al., 2019), in liquid, supercooled or mixed-phase conditions. The instrumental deployment including wind-tunnel, counter-flow virtual impactor, cloud droplet collectors allows to study the different phases of clouds separately.

Cloud droplets are efficient scavengers of chemicals resulting from the aerosol particles acting as cloud condensation nuclei and also from the dissolution of soluble species transferred from the gas phase. The liquid phase of clouds is a particularly reactive media, influencing the life-cycle of many key atmospheric compounds. Cloud water therefore contains a myriad of primary and secondary chemical compounds from both natural and anthropogenic origins that can react, producing additional secondary products that can return to the gas or aerosol phases upon cloud evaporation.

Cloud chemistry studies at puy de Dôme started early 2000 with direct measurements of the scavenging efficiencies of aerosol particles and the main organic and inorganic gases both for liquid and mixed-phase clouds (Laj et al., 2001; Sellegri et al., 2003a, 2003b; Voisin et al., 2000) These works demonstrated that equilibrium predicted by thermodynamics cannot explained the observed concentrations. These specific experiments were completed by :

- addressing a long-term variability approach in 2004 and 2014 (Deguillaume et al., 2014; Marinoni et al., 2004),
- the innovative dimension of biological processes (Amato et al., 2005),
- the recent use of high resolution mass spectrometry to address the complex organic molecular characterization (Bianco et al., 2018, 2019a).

The PUY station is, indeed, an ideal site to collect cloud samples over long time periods and to characterize them chemically (see Section 5.4.1). During their atmospheric transport, air masses arriving at PUY are enriched by chemical compounds emitted by various sources. This leads to a specific signature observed in the cloud water chemical composition. This long-term monitoring of the chemical composition of cloud water allows to classify clouds into various categories (highly marine, marine, continental and polluted) that serve to define a chemical scenario for modeling studies (Deguillaume et al., 2014). For this, multivariate statistical analyses were performed considering the concentrations of main inorganic ions and pH; the back-trajectories of the air masses that reach the PUY station for each cloud event were also calculated to confirm the classification (Bianco et al., 2018). This cloud database has been recently complemented by measurements of trace metals that help to evaluate the impact of anthropogenic and natural sources on the cloud and to better discriminate the origin of the air masses (Bianco et al., 2017).

In parallel to cloud chemical composition, microorganisms present in clouds (bacteria, fungi) are studied systematically at PUY since 2003 (Section 5.4.1). This long term observation of cloud biological composition is a unique database. Main sources of microorganisms are associated with continental emissions, more specifically vegetation. Few genera dominate the pool of cultivable microorganisms: for bacteria, for example, *Pseudomonas* and *Sphingomonas* (Proteobacteria) are frequently detected. The recurring presence of certain microbial genera reveals that vegetation is one of the major sources, which could also result from the development of strategies of specific aerosolisation or properties compatible with their survival in clouds (Joly et al., 2015; Vaïtilingom et al., 2012). Microbial concentrations reach values of roughly $10^4$-$10^5$ bacteria per milliliter and $10^3$-$10^4$ fungi and yeasts per milliliter in cloud water. Working directly on microorganisms isolated from cloud waters, it has been shown that they are able to synthetize molecules such as surfactants that modify the CCN activity (Renard et al., 2016) or siderophores that chelate iron in cloud waters and thus potentially impact the chemistry of the atmospheric aqueous phase (Passananti et al., 2016; Vinatier et al., 2016). The presence, abundance and variability of ice nucleation active biological particles and microorganisms were also studied at PUY and Opme stations (Joly et al., 2013, 2014; Pouzet et al., 2017). Biological ice nuclei are known to induce freezing at elevated temperature (>-10°C) (Joly et al., 2013) but their abundance in the atmosphere has been poorly investigated. The number of biological ice nuclei relative to the total number was estimated at 92% of samples between -6 and -8°C and at 65% at -10°C (Joly et al., 2014) for cloud water samples collected at PUY in contrasted environmental conditions. Recently, in order to better understand microbial life conditions in clouds and its eventual impacts, molecular studies were conducted. An important biodiversity including active microbial groups was depicted by high throughput sequencing (Amato et al., 2017). Their metabolic functioning was explored by metagenomics / metatranscriptomics approaches (Amato et al., 2019). The results demonstrated that

microorganisms face oxidants, osmotic shocks and cold in clouds, which potentially impacts cloud physics and chemistry by acting on the oxidant capacity, iron speciation and availability, and the carbon and nitrogen atmospheric budgets.

The cloud medium allows complex transformations of chemical compounds by both photochemical and biological processes. Respective efficiencies of those transformations working directly on the cloud medium have been intensively studied in this last decade. The oxidative capacity of cloud waters sampled at PUY has been investigated during three field campaigns from 2013 to 2014. It was demonstrated that hydroxyl radical (HO$^\bullet$) production in the aqueous phase is efficient and mainly due to the photolysis of hydrogen peroxide (Bianco et al., 2015). This confirms previous measurements at the PUY station where $H_2O_2$ was monitored (Marinoni et al., 2011). $H_2O_2$ showed diurnal variation, demonstrating its photo-reactivity, and also a dependence on the air mass origin. Abiotic pathways, such as HO$^\bullet$ mediated mechanisms, compete with biotic degradation. Microorganisms in cloud water are metabolically active and can metabolize organic compounds that are used as nutrients (Bianco et al., 2019b; Vaïtilingom et al., 2011, 2013). They are also able to degrade pollutants such as phenols (Lallement et al., 2018b). Moreover, they interact with reactive oxygen species, thus playing another role in cloud chemistry: they destroy oxidants that can potentially damage cells, reducing the concentration of precursors ($H_2O_2$ in particular) of reactive oxygen species (Vaïtilingom et al., 2013; Wirgot et al., 2017). However, the endogenous microflora was not shown to impact the steady state hydroxyl radical concentrations (Lallement et al., 2018a).

All those degradation processes need to be evaluated and compared in the frame of the cloud system. For this a new cloud chemistry model called CLEPS (Cloud Explicit Physico-Chemical Scheme) has been recently developed (Mouchel-Vallon et al., 2017) allowing the description of the oxidation of inorganic and organic compounds in the gas and aqueous phases, as well as the mass transfer between these two phases. This model has been recently evaluated towards the long term observations of the aqueous phase composition of the clouds sampled at PUY (Rose et al., 2018).

## 6.4 Water vapor and rain

The water is naturally present in the atmosphere in all three physical states: gas (water vapor) in the atmosphere, solid ice and snow, and liquid water (liquid clouds and rainfall). The atmospheric water vapor has a complex life cycle, which includes vertical and horizontal transport, mixing, condensation, precipitation and evapotranspiration. The understanding of cloud processes and precipitation from the local to synoptic scales involves a finely described multi-scale interaction between water vapor, cloud hydrometeors and precipitation, and their dynamics and radiative properties.

The GPS technology combined with meteorological measurements allows to continuously measure the integrated water vapor content in the atmosphere with a high temporal resolution. Having stations at different altitudes but within a limited spatial range to document the vertical variability of atmospheric water vapor has shown that the urban layer (i.e. the layer between the two different altitude sites Opme and Cézeaux) exhibits somewhat constant water vapor content, and the major water vapor variations occur in the upper troposphere level, in particular in the presence of westerly flows that bring elevated water vapor content over the mountain ridge (Van Baelen and Penide, 2009). GPS stations have been installed in Opme, PUY and Cézeaux, and are currently operating, allowing studies on the relationship between water vapor and rainfall

(Labbouz et al., 2013). A recent study based on 5 years of measurements of integrated water vapor content by GPS, mixing ratio of water vapor at the surface by humidity sensor and rainfall rate by rain gauges and disdrometers showed a statistical link between these various parameters, and more precisely that in 76% of the cases, a peak of water vapor precedes that of rain by 20 min. The wetter the atmosphere, the greater the precipitations and the temporal delay are important (Labbouz et al., 2015). An analysis of decadal *in-situ* and remote sensing observations of water vapor based on the measurements at CO-

PDD make it possible to document the variability, cycles and trends of surface and tropospheric water vapor at different time scales and the geophysical processes responsible for the water vapor distributions (Hadad et al., 2018), showing that the annual cycle of water vapor is clearly established for the two sites of different altitudes and for all types of measurements. Cézeaux and PUY present almost no diurnal cycle, suggesting that the variability of surface water vapor at this site is more influenced by a sporadic meteorological system than by regular diurnal variations. The vertical dimension given by the

LIDAR and GPS measurements showed that the LIDAR climatological profiles of water vapor present the same annual cycle but a larger variability than satellite profiles (COSMIC-FORMOSAT and AQUA-AIRS). The X band RADARs have been extensively used in international campaigns such as COPS and HYMEX to study the interaction of convective rain with water vapor, in conjunction with GPS observations (Labbouz et al., 2013; Planche et al., 2013; Van Baelen et al., 2011), or orography, in conjunction with MRR (Zwiebel et al., 2016). The local synergy of simultaneous X-band RADAR and rain

gauge measurements based on innovative geostatistical methods has also been used to improve the fine mapping of rainfall (Seck and Van Baelen, 2018). This geostatistical technique allowed, for example, an improvement of 54% in terms of bias reduction for kriging.

Finally, the concentration of ice nucleating particles in precipitation has been measured in rain samples at Opme during a more than one year period, showing variability over two orders of magnitude at a given temperature. The data support a

805 natural link with hydrological cycle, as well as a strong impact of human activities on the role of INP as triggers of precipitation (Pouzet et al., 2017).

**6.5 Dynamics and long range transport of radionuclides**

In the context of a changing climate, it is of primary importance to be able to detect and quantify changes in GHG, but also to connect these changes to the different atmospheric compartments in order to identify the dynamical links and the strength

of their exchanges. Jet streams are strong zonal winds in the upper troposphere which play an important role on the dynamical coupling between the stratosphere and troposphere, and which are also likely to induce long range transport of large quantities of atmospheric constituents (gas and aerosol). The analysis of the 15 years data of the Opme VHF profiler (1999-2014) compared with similar series at Lannemezan (Southern France), established the climatological behavior of tropospheric wind, which reveals dominant westerly high tropospheric winds with a maximum speed in winter of 22 m s$^{-1}$ at

the altitude 9 km (Baray et al., 2017). This predominance of the westerly direction is due to upper tropospheric jet streams, strong winds at upper tropospheric altitudes which, in the most intense cases, exceed 50 m s$^{-1}$, can be at the origin of stratosphere- troposphere exchanges. The jet stream shows a clear seasonality, with a predominance in winter (3 to 10% of

hourly profiles) and a minimum in summer (less than 1%), and a decadal trend (+ 1.6 ± 1.2% per decade). These results are corroborated by the analysis of radionuclides (Beryllium-Sodium report) and the analysis of the Lannemezan upper tropospheric wind data series.

The presence of radionuclides in aerosol and rain has also been investigated (Bourcier et al., 2012a). Measurements were conducted at three sampling sites located at different altitudes during two years, both in the rain and aerosol phases. The rain was sampled at Opme (boundary layer site) while the aerosol particles were collected at two different altitudes (Cézeaux and PUY), which allow a better characterization of the vertical atmospheric column being washed out. Various chemical analyses were performed during specific campaigns to characterize reactive ($NO_3^-$, $SO_4^{2-}$, $NH_4^+$ and $K^+$) and inert ([7]Be, [210]Pb and [137]Cs) species transfer from the aerosol to the rain phase. Using the classical washout ratio calculated with the aerosol concentration sampled at the same altitude than the rain collectors, we observed a seasonality of the washout ratio for radionuclides, with higher value in winter and lower value in summer. At PUY, local contamination does not influence the aerosol concentration.

Long-distance transport events of atmospheric constituents can be observed by instruments operating at the CO-PDD site. For example, Cesium isotopes ([134]Cs and [137]Cs) are directly related to nuclear accidents and their concentration in the cloud and rainy waters of PUY and Opme respectively increased by a factor of 40 during the Fukushima accident in March 2011 (Masson et al., 2015). Over the following weeks, concentrations decreased more slowly in cloud water than in rainwater, and more slowly in rainwater than in aerosol sampling. [134]Cs was detected in the aerosol phase, in the rain and in the cloud water for 3 months, 11 months and 18 months respectively, after the accident. Recent European-scale detection events of anthropogenic radionuclides ([131]I, [75]Se, [106]Ru) were observed in France thanks to the radionuclide monitoring performed in Opme and at PUY (Masson et al., 2015). Besides accident situations, routine comparison of the [137]Cs results at the summit and in Opme makes it possible to highlight the presence of sporadic high-altitude Saharan dust intrusions.

**7 Conclusions and future plans**

The puy de Dôme is a century-old meteorological observation site, and CO-PDD in its present configuration has now more than twenty years of microphysical, chemical and remote sensing measurements. CO-PDD is devoted to the long-term atmospheric survey in the context of climate change, and offers a unique instrumentation and dataset to document the complex connections linking precipitation and radiation, but also gases and particles clouds interactions. This allows CO-PDD to play an active role in European infrastructures such as ACTRIS or ICOS. Recent instrument acquisitions funded through regional, national, and European projects continue to strengthen the observation and research capacity of CO-PDD and will allow in the future to:

- continue to respond to the scientific air quality and climate change key questions, and to provide data available for the scientific community,
- develop new research activities, especially for linking the different atmospheric compartments.

For example, the multiphasic organic composition (MOCCA) instrument (PTR-ToF-MS by Ionicon) currently under development will help to understand the evolution of atmospheric particles as they are formed from different gas phase precursors, react and grow in the atmosphere, prior to being scavenged by cloud. This new platform will be operational at the puy de Dôme station in 2020 to document the gas/aerosol/cloud organic speciation. This multiphasic organic composition instrument (MOCCA) is composed of a PTR-ToF-MS 6000x2 (Ionicon) coupled with a fast GC to directly sample and

analyze the volatile organic compounds (VOC) (including biogenic VOC and Oxygenated VOC), with an aerosol inlet (CHARON), particle Inlet to directly sample and analyze the chemical composition of atmospheric sub-µm particulate organic matter. A separate head-space sampler is fitted to analyze cloud water samples.

The synergy of cloud and MRR RADARs will allow the monitoring of the cloud lifecycle and corresponding rain structure as well as to observe the cloud to rain transition with a high spatial and temporal resolution. Two hydrometeor RADAR

instruments will in 2019 or 2020 for the study of clouds and precipitation: a pulsed cloud RADAR at 35 Ghz and a vertical meteorological RADAR profiler at 24 Ghz to obtain the Doppler spectra of hydrometeors between 15 m and 6 km. These instruments will be first deployed at a site near the Aydat Lake area (12 km south of PUY) to study the microphysical and thermodynamical characteristics of precipitating clouds in relation to the biological and physico-chemical composition of cloud water, and linking these properties with the subsequent impact on terrestrial and aquatic ecosystems. Afterward, these

instruments will operate on a routine basis at the Opme or Cézeaux site and integrate in the framework of CO-PDD.

Finally, the characterization of bioaerosols which are potentially important to organic aerosols in the atmosphere, and play an active role as both cloud condensation nuclei and ice nuclei, providing us with a new means to understand their sources and links with the cloud system (as CCN for example). Biological aerosol particles are also important in the context of air quality studies and health effects research. Recently, a bioaerosol counter ("WIBS-NEO", DMT, USA) has been acquired.

This fluorescent aerosol particle size spectrometer has been built for real-time detection of bio-aerosols. This instrument will provide detailed information (detailed size, asymmetry factor) on atmospheric bacteria, molds, pollen and other bioaerosols. Three UV wavebands have been selected to optimize detection of common bioaerosols (tryptophan and NADH). It has been installed at the PUY station since October 2018 and data analysis is ongoing.

These new instruments will enable CO-PDD to be an important element of the French and European atmospheric research

landscapes in order to :

- respond to current scientific issues on processes characterization and long-term monitoring,
- maintain and develop the potentialities for hosting  external teams (transnational access),
- continue to export the instrumental expertise on other sites (e.g., *in-situ* measurements of the Maïdo station, Indian Ocean, southern subtropics),

- constitute an efficient tool for early detection of atmospheric hazards (*e.g.,* pollution peaks, volcanic eruptions, forest fires, nuclear accidents).

**Acknowledgments**

CO-PDD is an instrumented site of OPGC observatory and LaMP laboratory, supported by the Université Clermont Auvergne (UCA), by Centre National de la Recherche Scientifique (CNRS-INSU) and by the Centre National d'Etudes
Spatiales (CNES). In addition, some instruments and operations have been founded by Institut de Radioprotection et de Sûreté Nucléaire (IRSN) and Commissariat à l'Energie Atomique et aux énergies alternatives (CEA), Région Auvergne and Auvergne-Rhône-Alpes, Conseil Départemental 63, Clermont Métropole. The research activities have received funding from the European Union's Horizon 2020 research and innovation programme under grant agreement No 654109 (ACTRIS2 – H2020). The authors also strongly acknowledge the financial support from Fédération des Recherches en Environnement
through the CPER founded by Region Auvergne - Rhône-Alpes, the French ministry, ACTRIS Research Infrastructure and FEDER European Regional funds. Some research projects funded by the French national research agency (ANR) also contributed to the data production of CO-PDD (CHAIN ANR-14-CE01-0003, BIOCAP ANR-13-BS06-0004…).

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

**Table 1: List of instruments deployed at PUY.**

| Instrument | Parameter | Measurement period | Typical temporal resolution | Network, Research infrastructure, national and local services… | Laboratory involved (other than LaMP or OPGC) |
|---|---|---|---|---|---|
| Sonic Anemometer | Wind speed and direction | 1995-present | 5 min | | |
| Meteorological station | Pressure, temperature, humidity | 1995-present | 5 min | | |
| ACSM (Aerodyne) | Aerosol chemical composition | 2015-present | 10 min | CLAP, ACTRIS, GAW | |
| MAAP | Black carbon | 2000-present | 5 min | CLAP, ACTRIS, GAW | |
| CPC (TSI) | Total aerosol number | 2000-present | 5 min | CLAP, ACTRIS, GAW | |
| SMPS, OPC Grimm | Aerosol size distribution | 2001-present | 5 min | CLAP, ACTRIS, GAW | |
| Photometer | Aerosol optical depth | 2005-present | 1 hour | CLAP, ACTRIS, GAW | |
| TEOM | Aerosol mass | 2005-present | 2 min | CLAP, ACTRIS, GAW | |
| (N)AIS | Aerosol (charged and/or neutral) size distribution | 2006-2015 | 5 min | CLAP, ACTRIS, GAW | |
| PSM | Cluster particles size distribution | 2011-2014 | 1-4 min | | |
| HTDMA (homemade) | Aerosol hygroscopicity | 2008-2012 | 3 hours | ACTRIS | |
| Scanotron (homemade) | CCN | 2012-present | 10 min | ACTRIS | |
| Nephelometer (TSI) | Scattering coefficient | 2003-present | 5 min | CLAP, ACTRIS, GAW | |

| Instrument | Species | Period | Frequency | Network | Institution |
|---|---|---|---|---|---|
| Filter | Radionuclide aerosols | 2005-present | 1 week | OPERA | IRSN |
| Alpha spectrometer | Radon-222 | 2002-present | 2 hours | RAMCES | LSCE, Paris |
| Fluorescence UV (TEI 43 CTL) | $SO_2$ | 1995-present | 5 min | PUY-GAZ, EMEP | |
| UV absorption (TEI 49 i) | $O_3$ | 1995-present | 5 min | PUY-GAZ, EMEP, GAW | |
| Infra-Red absorption (TEI) | CO | 2002-2012 | 5 min | PUY-GAZ, EMEP, ICOS | |
| $O_3$ chemiluminescence (TEI 42 CTL) | $NO_x$ | 2003-present | 5 min | PUY-GAZ, EMEP, ACTRIS, GAW | |
| GC-MS (Perkin Elmer) | VOC (NMHCs, BVOC) | 2010-2016 2017-present | On campaign Once a week | PUY-GAZ, ACTRIS, GAW | |
| NDIR (Licor) | $CO_2$ | 2000-2012 | 5 min | RAMCES | LSCE, Paris |
| CRDS (Picarro) | $CO_2$, $CH_4$, CO | 2012-2015 | 5 min | RAMCES, ICOS | LSCE, Paris |
| GC-FID/ECD (Agilent) | $CO_2$, $CH_4$, $N_2O$, $SF_6$ | 2010-2015 | 5 min | RAMCES, ICOS | LSCE, Paris |
| GC-FID (Agilent) | NMHCs | 2018-present | 2 hours | PUY-GAZ, ACTRIS, GAW | |
| Cloud Impactor (Homemade) | Cloud microorganisms | 2003-present | 2 hours | PUY-CLOUD | ICCF |
| Hi-volume impingers (Kärcher DS6) | Microorganisms and biological particles in clouds and aerosols | 2014-present | 2 hours | PUY-CLOUD | ICCF |
| Impactor | Radionuclide cloud | 2007-present | 1 week | OPERA | IRSN |
| GPS | Integrated Water Vapor | 2011-present | 1 hour | RENAG | |

**Table 2: List of instruments deployed at Opme.**

| Instrument | Parameter | Measurement period | Typical temporal resolution | Network, Research infrastructure, national and local services… | Laboratory involved (other than LaMP or OPGC) |
|---|---|---|---|---|---|
| Meteorological station | Pressure, Temperature, Humidity | 2017-present | 5 min | | |
| Rain gauge | Precipitation | 1998-present | 5 min | | |
| Disdrometer | Precipitations | 2006-present | 5 min | | |
| VHF profiler | Wind profiles | 1998-2015 | 15 min | E-Profile | |
| | Radionuclide aerosols | 2004-present | 1 week | OPERA | IRSN |
| | Radionuclide rain | 2004-present | 1 month | OPERA | IRSN |
| MRR | Rain reflectivity profiles | 2006-present | 5 min | | |
| Precipitation collector | Biological ice nucleating particles, chemical composition | 2015-present | 1 day | | ICCF |
| GPS | Integrated Water Vapor | 2011-present | 1 hour | RENAG | |


**Table 3: List of instruments deployed at Cézeaux and Aulnat.**

| Instrument | Parameter | Measurement period | Typical temporal resolution | Network, Research infrastructure, national and local services… | Laboratory involved (other than LaMP or OPGC) |
|---|---|---|---|---|---|
| Meteorological station | Pressure, Temperature, Humidity | 2002-present | 5 min | | |
| Rain gauge | Precipitations | 1994-present | 5 min | | |
| MRR | Rain reflectivity profiles | 2014-present | 5 min | | |
| Band X RADAR | Rain reflectivity | 2006-present | 15 min | | |
| GPS | Integrated Water Vapor | 2007-present | 1 hour | RENAG | ENSTA Bretagne |
| Disdrometer | Precipitations | 2007-present | 5 min | | |
| LIDAR | Aerosol/$H_2O$ profiles, cirrus | 2008-present | From 1 min to 1 day | EARLINET, ACTRIS | |
| UHF profiler | Wind profiles | 2014-present | 15 min | E-Profile | LA Toulouse |
| Full sky imager | Cloud fraction | 2015-present | 10 min | | |

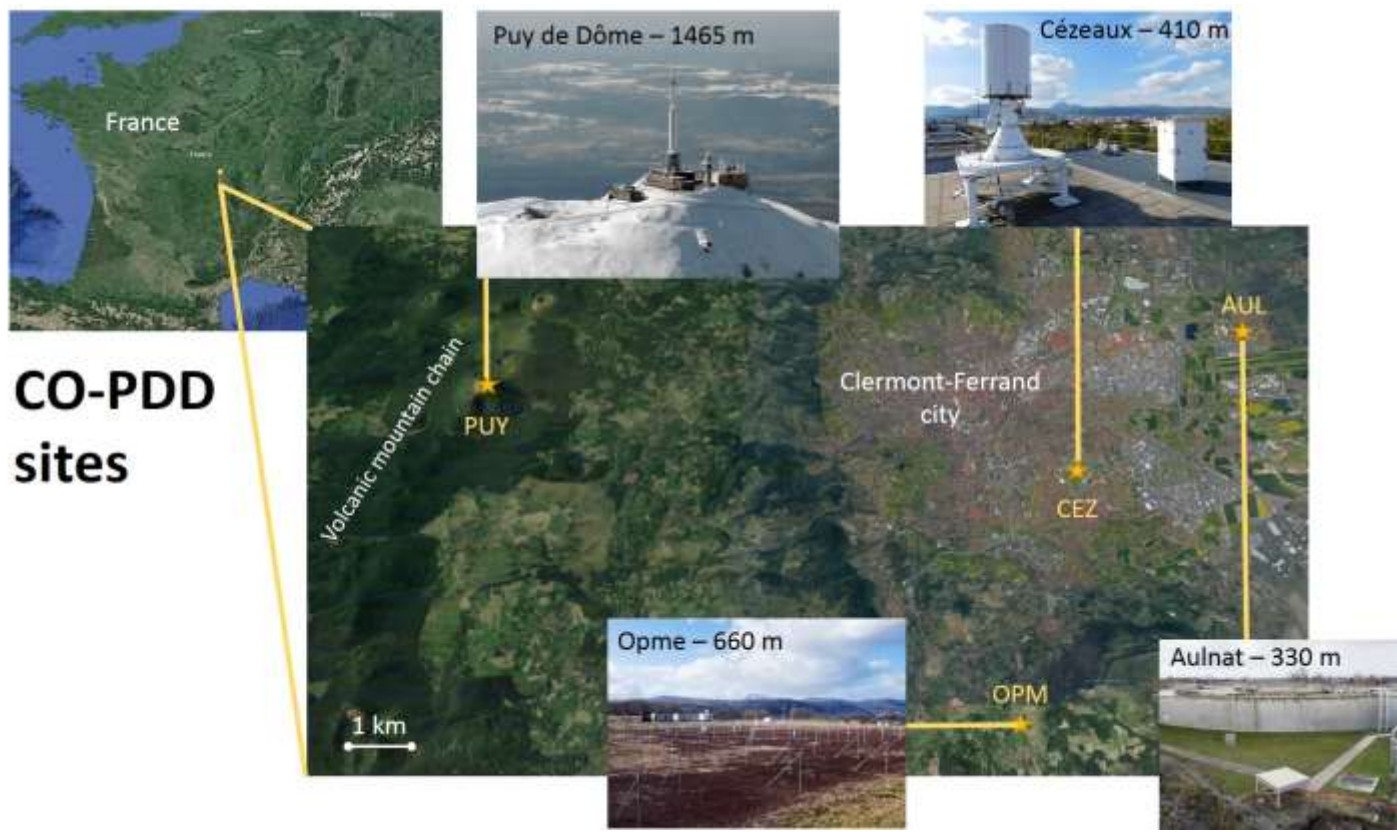

Figure 1: Overview maps showing the location and photos of the CO-PDD sites. This figure was created using ©Google Map.


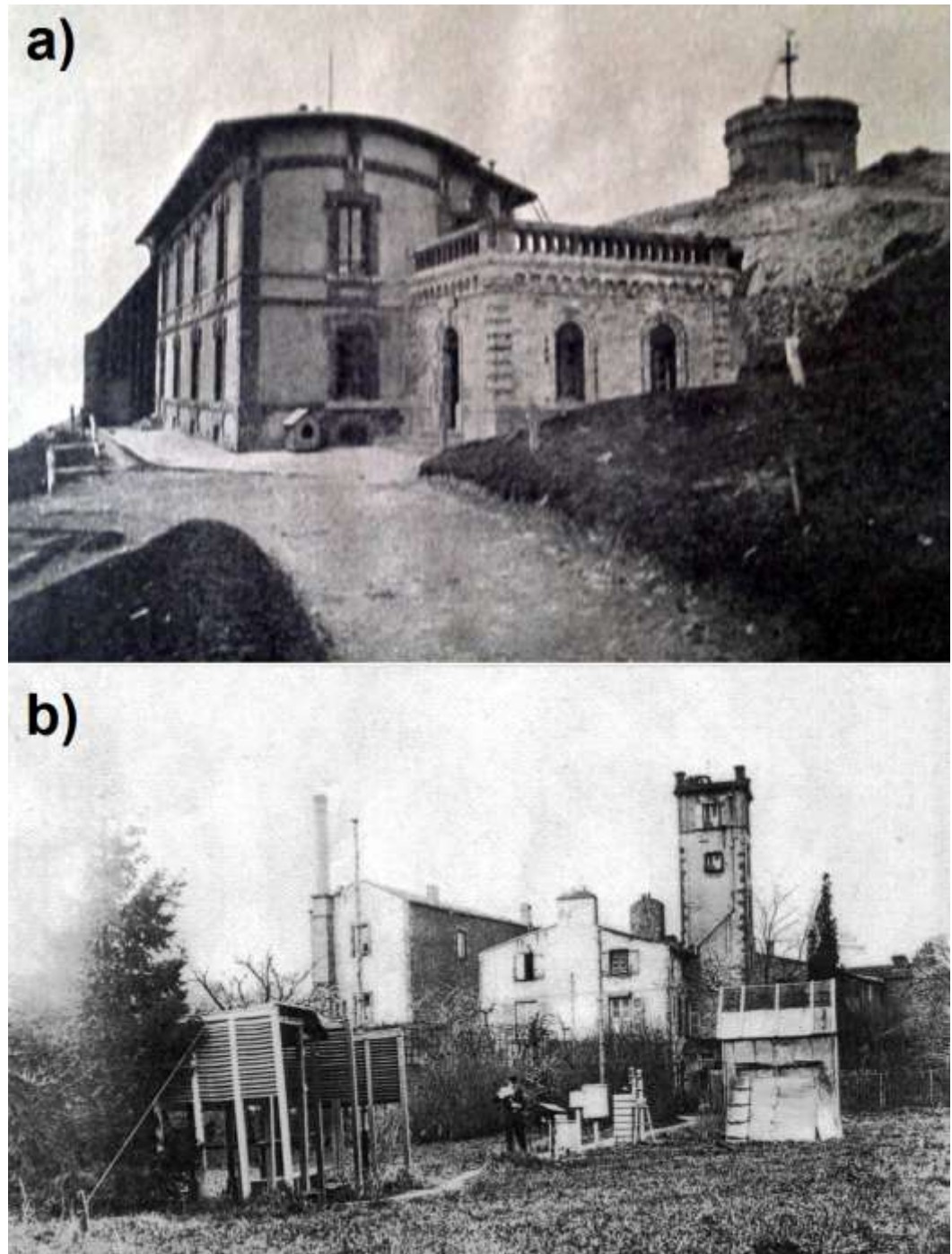

**Figure 2: Historical views of the observatory. (a) Living chalet-style house in front of the observation tower at the top of the puy de Dôme volcano, (b) lowland station in the center of Clermont-Ferrand, rue Rabanesse.**

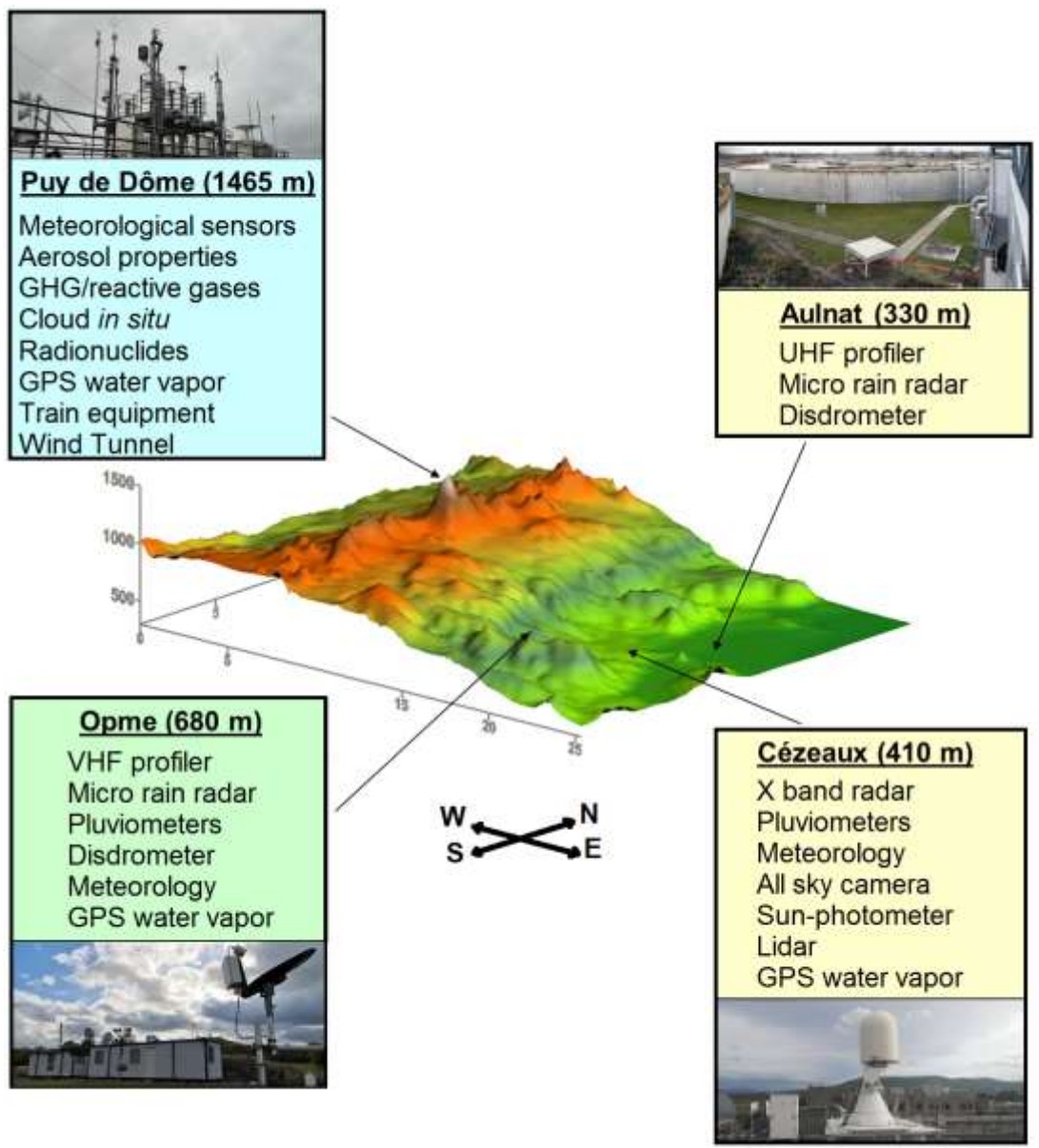

**Figure 3: Instruments and measurement sites of the CO-PDD atmospheric research station.**

**Figure 3: CO-PDD parameters or instruments and relative sites.**

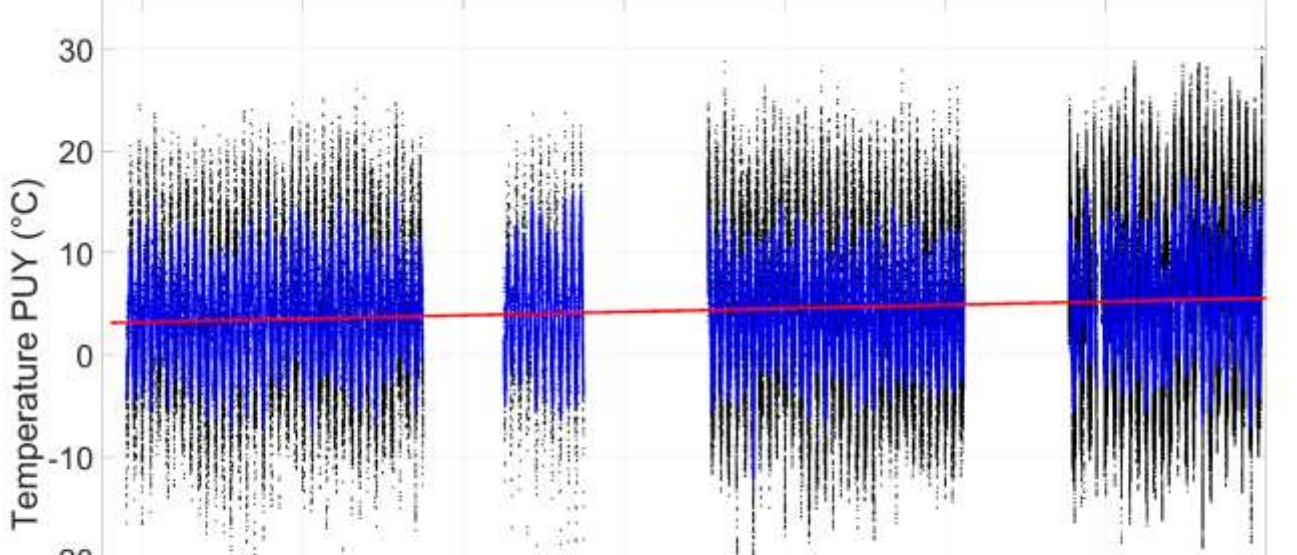

**Figure 4: Long series of temperature at PUY station. The hourly means are in black, the monthly means are in blue and the linear regression line is in red (slope: +1.4 ± 0.7 °C per century).**

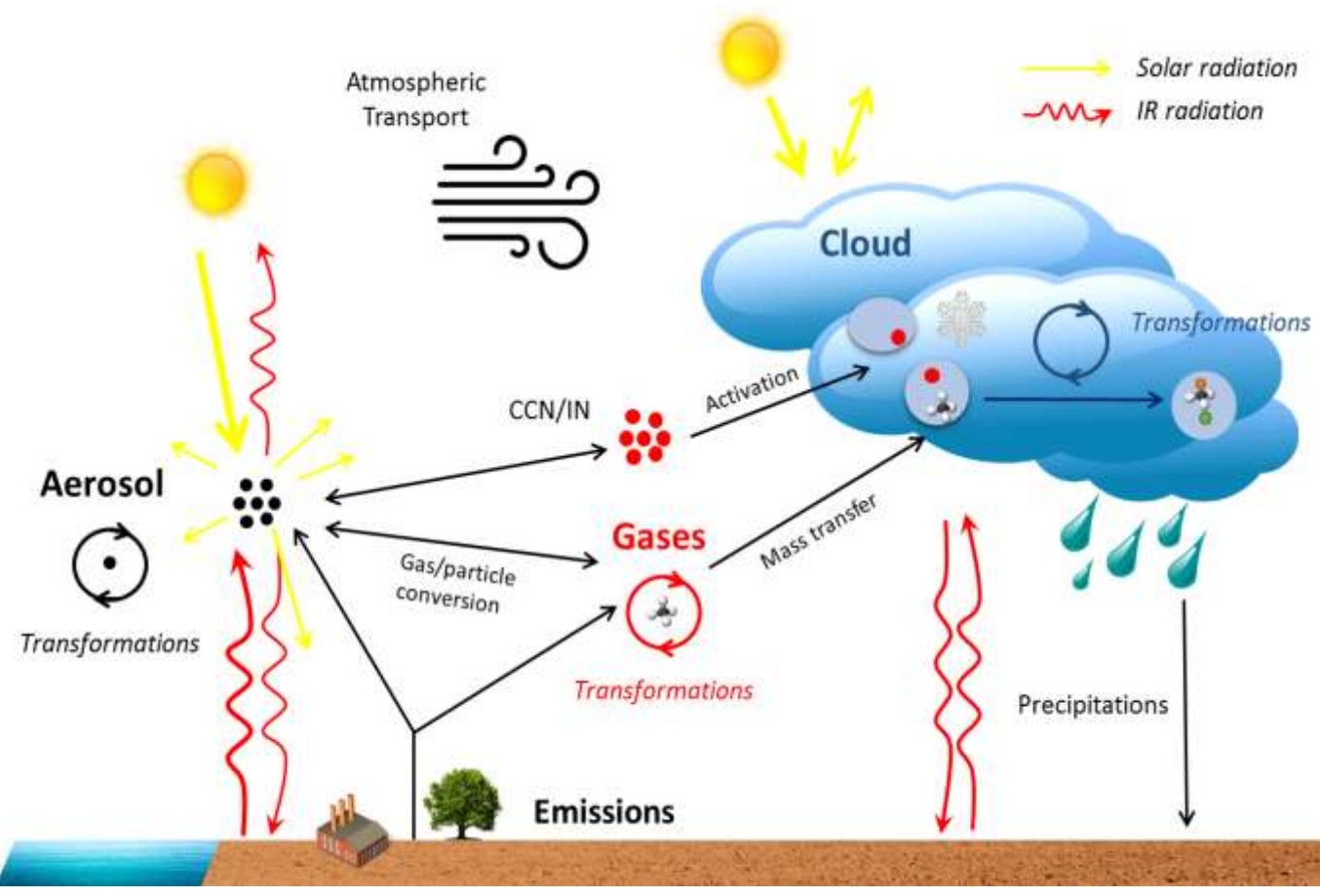


**Figure 5: Schematic representation of atmospheric processes linking aerosol particles, clouds, precipitation, and radiation.**

## a) INLET

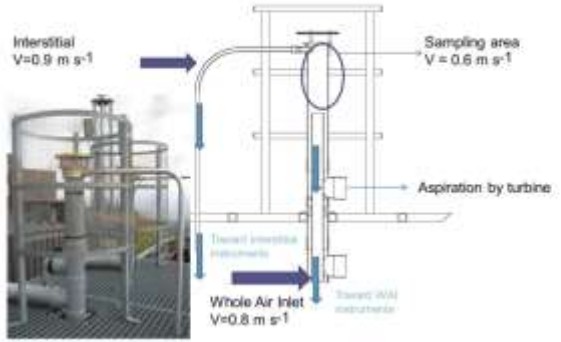

## b) AEROVOCC

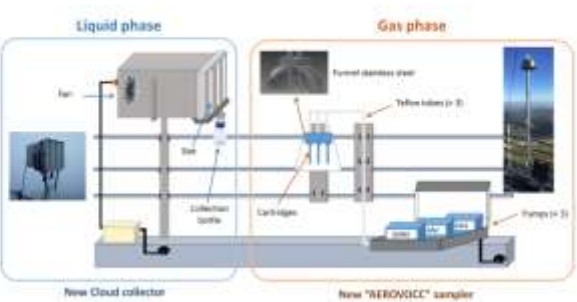

## c) TRAIN INSTRUMENTATION

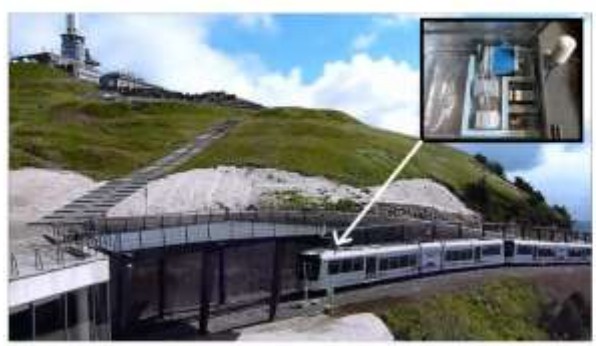

## d) LIDAR

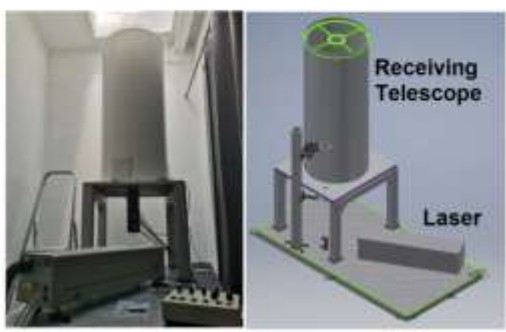

## e) WIND TUNNEL

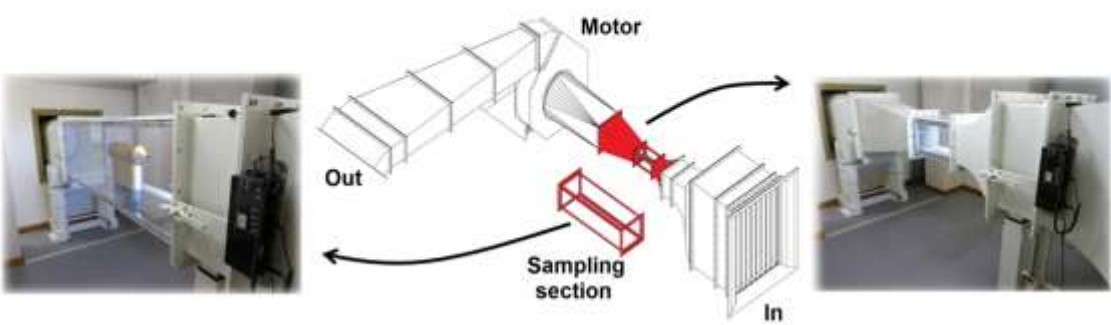

**Figure 6: Examples of instrumental development. (a) Inlet for gas and aerosol sampling, (b) AEROVOCC : *in-situ* multiphasic VOC sampling system, (c) Tourist train instrumentation, (d) LIDAR COPlid development and (e) wind tunnel.**

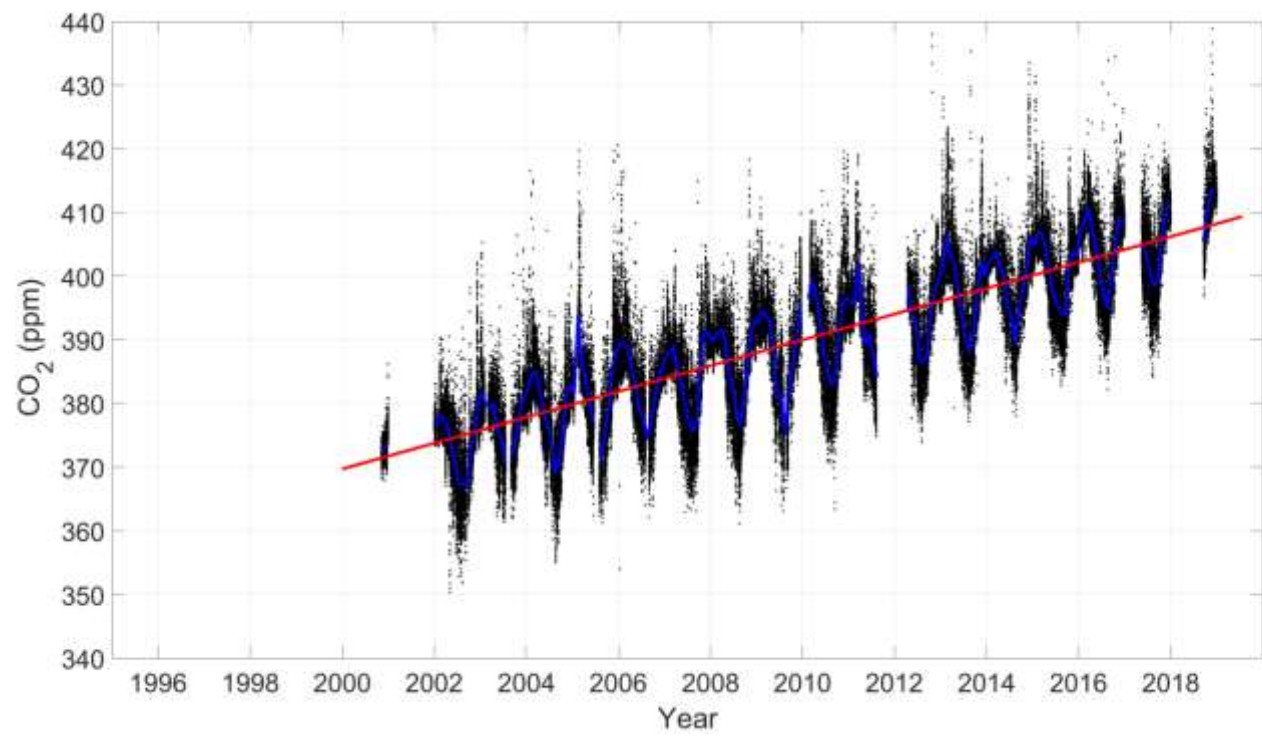

**Figure 7: Long series of $CO_2$ at PUY station (slope: +20.2 ± 1.7 ppm per decade). The hourly means are in black, the monthly means are in blue and the linear regression line is in red.**

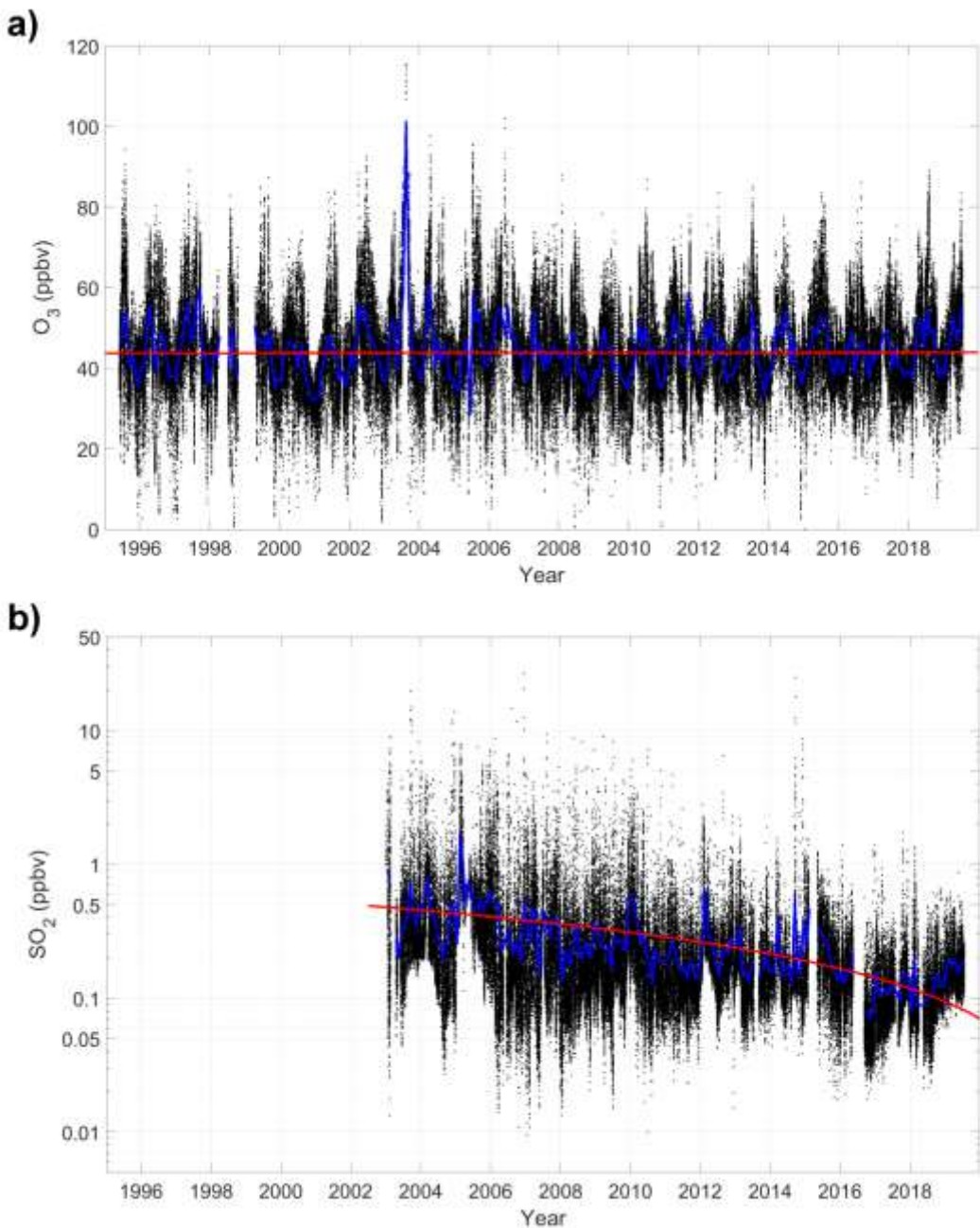

**Figure 8: Long series of reactive gases at PUY station: O$_3$ (a, slope: -0.1 ± 1.2 ppbv per decade) and SO$_2$ (b, slope: -0.23 ± 0.05 ppbv per decade). The SO$_2$ values have been plotted on a logarithmic y axis. The color code is the same as Figures 4 and 7.**

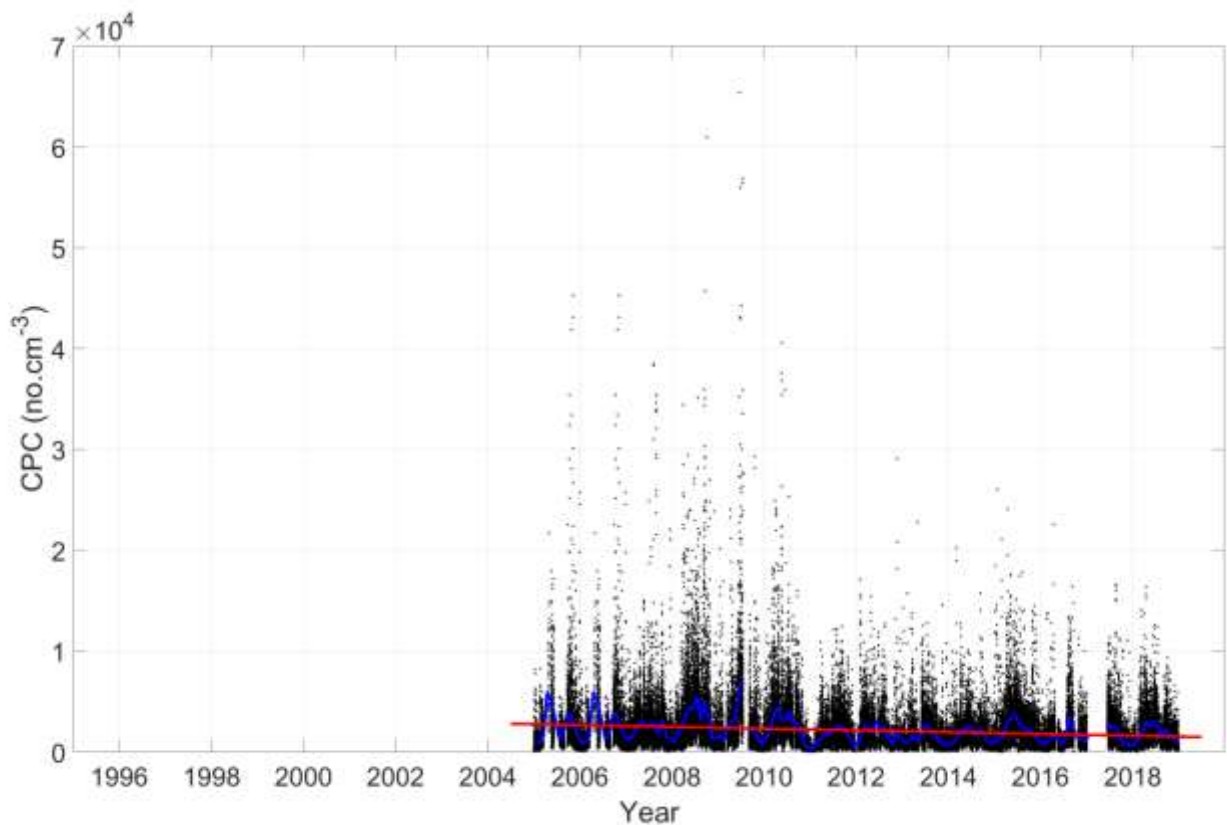

**Figure 9: Long series of aerosol particles number at PUY station (slope: -9 ± 5 10² cm⁻³ per decade). The color code is the same as Figure 4, 7 and 8.**

**Appendix A : Data availability**

All data produced by CO-PDD are effectively accessible free-of-charge to a wide user community. Because CO-PDD is part of different national and international initiatives, including the European Research Infrastructures ACTRIS and ICOS, with specific data policies, or the international networks AERONET, not all CO-PDD information can be retrieved by a single data hub, but are split between local, national and international data centers. Since 2020, a strong effort is made to apply FAIR (Findable, Accessible, Interoperable and Reusable) principles to some data produced at CO-PDD through the work done within the ENVRI-FAIR (H2020, 2020-2024) EU project that will facilitate uptake of data from different sources.

All data produced at CO-PDD are first stored on local servers including raw data. For optimizing their visibility and local use, some quicklooks, descriptions of measurement systems and near real-time data are accessible through the current OPGC internet portal (http://wwwobs.univ-bpclermont.fr/SO/mesures/direct.php). A new website with virtual observatory (http://wwwobs.univ-bpclermont.fr/vobs/index.php) allowing easier information on data is currently being developed.

A second layer of dissemination is the national level. CO-PDD provides all data produced within national and international initiatives to the national Earth Science Data System and its specific atmospheric component AERIS (https://www.aeris-data.fr/). AERIS centralizes all information besides the GPS water vapor column measurements measured at PUY, Cézeaux and Opme which are accessible through the RESIF-RENAG databases (http://renag.resif.fr and http://rgp.ign.fr) and besides the radionucleides measurements available through the OPERA data base (https://www.mesure-radioactivite.fr/en#/expert) upon request. In addition, some additional information of reactive gases can be accessed at the PAES portal (http://paes.aero.obs-mip.fr).

Finally, the 3rd layer of dissemination is the international level. Data is preserved for long-term archiving by the corresponding Research Infrastructures or networks as part of the archival process and in order to ensure good data management. Each step of the data lifecycle is documented in Research Infrastructures Data Management Plans, including collection, curation, data production, preservation, publishing and use of data. More specifically:

- ACTRIS related information are contained in ACTRIS-DC including in-situ ground-based measurements of aerosol properties and reactive gases (EBAS http://ebas.nilu.no/Default.aspx), the aerosol profiling (EARLINET https://earlinet.org including the EARLINET SCC (Single Calculus Chain) for additional lidar products). GAW and EMEP related information are also accessible through the EBAS data center.
- AERONET related information from sun-photometer measurements are available through the NASA Goddard Space Flight Center (http://aeronet.gsfc.nasa.gov)
- ICOS related information are accessible through the ICOS Carbon Portal (https://data.icos-cp.eu/)
- E-Profile database contains the Opme and Aulnat wind profiler measurements (http://eumetnet.eu/activities/observations-programme/current-activities/e-profile/radar-wind-profilers/)

**Appendix B : List of acronyms**

ACSM : Aerosol Chemical Speciation Monitor

ACTRIS : Aerosol Cloud and Trace gases Research Infrastructure

AERONET : Aerosol Robotic Network

AIS : Air Ion Spectrometer

AQUA-AIRS : Atmospheric Infrared Sounder

BC : Black Carbon

BIOCAP : Impact biologiques et photochimiques sur la capacité oxydante du nuage

CCNc : Cloud Condensation Nuclei chamber

CHAIN : Characterisation of atmospheric ice nuclei

CLAP : CLimate Relevant Aerosol Properties

CLEPS : Cloud Explicit Physico Chemical Scheme

CNRS : Centre National de la Recherche Scientifique

CO-PDD : Cézeaux-Aulnat-Opme-Puy De Dôme

COSMIC –FORMOSAT : Constellation Observing System for Meteorology, Ionospheric, and Climate

CPC : Condensation Particle Counter

CRDS : Cavity Ring-Down Spectroscopy

DMA : Differential Mobility Analyzer

EARLINET : European Aerosol Research Lidar Network

ECMWF : European Centre for Medium-Range Weather Forecasts

ENSTA : École nationale supérieure de techniques avancées (Brest)

EUSAR : European Supersites for Atmospheric Research

GAW : Global Atmosphere Watch

GC-FID : Gas Chromatography – Flame Ionization Detector

GCMS : Gas Chromatography Mass Spectrometry

GHG : GreenHouse Gases

GLONASS : GLObal NAvigation Satellite System

GPS : Global Positioning System

HTDMA : Hygroscopic Tandem Differential Mobility Analyser

HYMEX : HYdrological cycle in Mediterranean EXperiment

ICOS : Integrated Carbo Observation System

INSU : Institut National des Sciences de l'Univers

IRSN : Institut de Radioprotection et de Sûreté Nucléaire

JMA : Japan Meteorological Agency

KNMI : Royal Netherlands Meteorological Institute

LA  : Laboratoire d'Aérologie (Toulouse)

LaMP : Laboratoire de Météorologie Physique (Clermont Ferrand)

LIDAR : Light Detection and Ranging

LSCE : Laboratoire des Sciences du Climat et de l'Environnement (Paris)

MAAP : Multiangle Absorption Photometer

MRR : Micro Rain RADAR

NAIS : Neutral cluster and Air Ion Spectrometer

NASA : National Aeronautics and Space Administration

NOAA : National Oceanic and Atmospheric Administration

NTP : Normal Temperature and Pressure

OPC : Optical Particle Counter

OPERA : Permanent observatory of the radioactivity

OSU : Observatoire des Sciences de l'Univers

OPGC : Observatoire de Physique du Globe de Clermont Ferrand

PSM : Particle Size Magnifier

PTR-TOF-MS : Proton Transfer. Reaction - Time of Flight - Mass Spectrometer

PVM : Particule Volume Monitor

RADAR : Radio Detection and Ranging

RAMCES : Réseau Atmosphérique de Mesure des Composés à Effet de Serre

RENAG : REseau NAtional GNSS permanent

SMPS : Scanning Mobility Particle Sizer

TEOM-FDMS : Tapered Element Oscillating Microbalance-Filter Dynamic Measurement System

UHF : Ultra High Frequency

VHF : Very High Frequency

WAI : Whole Air Inlet

WIBS-NEO : Wideband Integrated Bioaerosol Sensor


## Appendix C : Encountered air masses

Puy de Dôme is the first mountain chain making an orographic barrier to the prevailing westerly winds. A long term monitoring of the chemical composition of clouds performed between 2001 and 2011, coupled with a back-trajectory analysis shown that, under cloudy situation, air masses reaching PUY mainly come from the West, North/West sectors and are classified as "marine" or "highly marine types" (Deguillaume et al., 2014).

In order to provide a more detailed description of encountered air masses arriving on the PUY site, including all weather conditions, we performed a statistical analysis of back-trajectories with the CAT model. CAT (Computing Atmospheric Trajectory Tool) is a recent evolution of the LACYTRAJ model (Clain et al., 2010). LACYTRAJ was a 3D kinematic trajectory code using initialization wind fields from the ECMWF ERA-Interim re-analyse with a horizontal resolution of 1° in latitude and longitude, and 37 vertical levels. A cluster of starting backtrajectory points was defined by the user and advected by the model using a bilinear interpolation for horizontal wind fields and time and a log-linear interpolation for vertical wind field, with a time resolution of 15 minutes between two trajectory points. The improvements from LACYTRAJ to CAT are:

- The initialization with wind fields from the recent ERA-5 ECMWF reanalysis of any temporal and spatial resolution. The adaptation to assimilate wind fields on a smaller scale from WRF and MESO-NH mesoscale models is currently under development.
- The integration of topography at a resolution of around 10 kilometers (Bezdek and Sebera, 2013).
- The possibility of automatically calculating forward and backward trajectories starting at any time, and not only on the hours of the ERA-Interim matrices (0, 6, 12, 18 UT), and with variable temporal resolution and starting pressure level (by example measured at an altitude station).

A statistical analysis of backtrajectories was then made using the CAT model for the years 2015 and 2016. Twice a day, a set of trajectories was calculated over a period of 24 hours from an area of 10 km around the puy de Dôme, and starting from the pressure level measured at the top of the PUY station. Figure C1 shows the total number of path points per 0.5° square and table C1 provides the percentage of backtrajectory points by sector  (for the full year, and by season).

We observe that the air masses arriving at PUY come mainly from the Atlantic Ocean. These westerly origins are mainly observed in winter (more than 50%), but they also occurs during the other seasons (more than 40%). 15 to 24% of air masses come from potentially polluted areas (easterly), with the larger values in spring (24%).

It may happen that desert dust aerosol particles are transported over long distance and are observed by the CO-PDD instrumental devices (mainly in winter), but with a transport time larger than 24 hours.

Finally, we observe that the category "near" is larger in summer. This may be due to the fact that the site is more frequently in boundary layer condition in summer, with therefore a higher local impact. An analysis using mesoscale model is necessary to investigate this point.

**Table C.1: Percentage of backtrajectory points in each sector, for the 2015-2016 period, whole year and per season. The category Near contains backtrajectory points in a 20 km radius circle around the PUY station.**

| Sector | 2015-2016 | Winter : DJF | Spring : MAM | Summer : JJA | Fall : SON |
|--------|-----------|--------------|--------------|--------------|------------|
| Near | 4,7 | 3,9 | 3,9 | 6,0 | 4,6 |
| SSW | 7,3 | 8,8 | 5,9 | 6,5 | 7,3 |
| SWW | 22,9 | 28,1 | 17,5 | 23,9 | 24,8 |
| NWW | 23,4 | 27,2 | 23,1 | 24,4 | 23,5 |
| NNW | 10,4 | 6,7 | 11,0 | 12,6 | 10,3 |
| NNE | 8,4 | 6,1 | 11,5 | 7,8 | 7,5 |
| NEE | 10,7 | 10,5 | 12,5 | 8,1 | 9,8 |
| SEE | 4,8 | 3,0 | 6,0 | 4,1 | 5,0 |
| SSE | 7,4 | 5,7 | 8,6 | 6,6 | 7,2 |

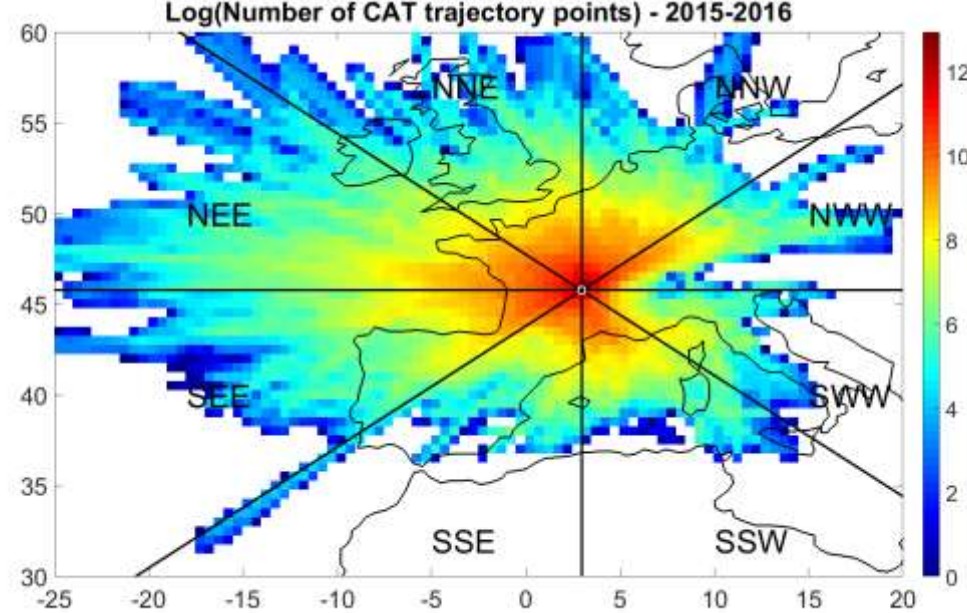

**Figure C.1: Logarithm of the number of back-trajectories points arriving at the summit of the PUY station for the 2015-2016 whole period. The white circle has a radius of 20 km and illustrates the category "Near" of Table C1, and the black lines separates the different sectors.**