# Peer review of "Cézeaux-Aulnat-Opme-Puy De Dôme: a multi-site for the long term survey of the tropospheric composition and climate change"

_Atmospheric Measurement Techniques, 2019_

## Referee Comment (RC1) · Anonymous Referee #1 · 3 Jan 2020

The paper summarizes long term measurements at a cluster of sites around the puy de Dome near Clermont Ferrand in Central France. It describes the geography of the site, the (history of) deployments of various instruments and the measurement techniques used. It synthesizes major results obtained from the observations and places them into the context of other EU sites. In my mind, the paper is interesting to publish in AMT, because it gives a good review of observations available at this internationally recognized site. This will be useful for the AMT readership. Before publication, the authors should take into account several general and specific remarks:

[Figure]

1) The data availability is generally not discussed. This is not in line with current practice, and it is even more required for this review paper which has among its aims to better document the available data sets for a more widespread use. The procedure to download available ta sets has to be given.

2) A section should be added which shows the type of encountered air masses at the sites, and in particular the PUY site. What are the geographic origins, which part is free tropospheric and from the boundary layer. How does local pollution from Clermont Ferrand affect the site. Are there any small scale meteorological studies adressing this question ? Part of information is given in the results sections, but should be grouped in a section prior to it.

3) For all instruments and geophysical variables, information about measurement periods and frequencies, detection limites and uncertainties should be given in synthetic tables.

4) Carefull rereading of the paper is still required. I have corrected several grammer errors, but surely not all.

5) Specific remarks :

P6li190 Âń 5 Technical description of observation systems Âż At the beginning of this paper I would like to see a small overview of the measurement gements, roughly corresponding to the titles of the sub-sections.

P6li193 Âń After preliminary . . .. Âż In this paragraphe, it seems implicitly that aerosol is mainly sampled in cloudy air masses. The sampling strategy needs further explanation.

P8Li255 Are there calibration procedures for ACSM or OC/EC ?

P12l370 VOC measurements What are the storage times for the cartridges, and what are the precautions for sample stability ?

P18li561 6.1 Trace gases I think, it would be better to regroup the GHG measurements

mentionned in the beginning and the end of the section. For each result on tendencies and seasonal variations, it would be interesting to state the particularity of results at PUY (if there ary any) with respect to other European sites.

The impact of local and remote sources on trends and variability should be stated, this is related to the height of the boundary layer. For which source area is this recptor site representative ?

P18, li574 Âń …., while its primary precursors like NOx has significantly decreased (not shown). Âż This implies that the seasonal variation should become less marked over time, due to less phochemisatry in summer and less titration in winter. Is that what the autors want to state? Can this be derived from figure 8 (from visual quick inspection : might be tru, but not sure whether statistically significant?

Âń This feature is consistent with the ozone trend reported at surface stations within the EMEP network between 1990 and 2012. Âż What feature is meant here ?

P18li583 Âń The analysis of the 3-years atmospheric time series revealed how the planetary boundary layer height drives the concentrations observed at PUY. Radionuclide measurements are used to determine the boundary layer / free tropospheric conditions. The CO2 surface flux are estimated and revealed a clear seasonalt cycle, under the influence of plant assimilation, and burning of fossil fuel. Âż

For these Âń one sentence statements Âż about results, are there publications and reports available ?

P19Li594 : Âń The variability of atmospheric aerosol at PUY shows a marked seasonal variation with a maximum during the summer, Âż Could the authors be more precise on the aerosol property concerned : PM mass, of which diameter, specific compounds, or particle number ?

P19Li594 : Âń as well as a non-negligible impact of nucleation (or new particle formation (NPF) events) on the particle size distribution Âń This argument should be apart

from the previous one, probably about aerosol mass.

Page19li604 : Âńcoupled with the atmospheric dynamics captured by the LIDAR located at Cézeaux Âż I think the above peri-phrase can be deleted, because there is no clue about what lidar mesaurements contributed to the analysis. Else, please explain more the of lidar measurements.

P20li631 : Âń In particular, the general trend is a significant decrease of the daytime aerosol diffusion coefficient during summer and autumn (Pandolfi et al., 2018). Âż It is not clear in this sentence, that a long term evolution is meant. What is the time scale ? P20li634 : Âń This trend is in line with aerosol mass (PM) decay observed at European level, in relation to SO2 emission regulations. Âż What is then the contribution of sulfate to PM ?

P20li634 : Âń and higher elevation sites or ocean-influenced air masses contain a larger fraction of highly hygroscopic aerosols (sea salts). Âż This is not clear, so for PUY is the larger hygroscopicity related to oceanic air mass origin.

P22LI689 : Âń This long term observation of cloud biological composition is a unique worldwide database. Âż Where are these data (and others) available ?

P23LI736 : Âńthat the urban layer Âż the presence of such a layer should be explained much earlier, in a section about the air mass origin of the observations.

P24LI764 : Âńatmospheric compartments in order to identify the dynamical links and the strength of their exchanges Âż Are there Rn measurements that could be exploited to detect the PBL origin of air masses ?

P24LI780 : Âńcharacterize reactive (NO3 -, SO4 2-, NH4 + and K+) …. species transfer Âż Are there systematic measurments of rain water composition, which allow documenting their long term evolution ?

P24LI783 : Âńwe observed a seasonality of the washout ratio for radionuclides, with higher value in winter and lower value in summer. Âż A possible reason for this ?

P37-40, tables 1,2,3 : Âń Laboratory Âż It should be something like Âń if other than LAMP or OPGG or so. The tables should contain in additionmore some more specifications : time periods and frequency of measurements (not just starting date), detection limit and uncertainty, site where data base is available.

Please also note the supplement to this comment:
https://www.atmos-meas-tech-discuss.net/amt-2019-383/amt-2019-383-RC1-supplement.pdf

---

## Referee Comment (RC2) · Anonymous Referee #2 · 24 Feb 2020

Review of Baray et al.,

Cézeaux-Aulnat-Opme-Puy De Dôme: A multi-site for the long term survey of the tropospheric composition and climate change

The article summarizes instrumentation, research and conceptual facility design built up at and around the Puy-du-Dome, central France, over the last decades, with historical roots reaching back to more than a century ago. Development, operations and future perspectives of an excellently equipped and integrated observatory are described

and main findings shortly pointed out or cited, but often only mentioned. The site's relevance emerges from its frequent use by the scientific community for process dedicated campaigns, field deployment of new measurement techniques and strategies as well as its and the contributing institutions' important roles in national and international research infrastructures.

The description of the CAO-PDD observatories, their relevance, concepts, aims and integration is comprehensive and useful to AMT readers. A broad selection of results, including many references, convinces that excellent scientific results have been inferred from the CAO-PDD measurements. A review like this may, for brevity and clarity, discuss part of the results qualitatively, however, its added value develops from their combination and their synopsis. I'm missing a number of important figures either in the text, as table(s) or as plots in order to serve as a 'first stop' also for external readers aside the European atmospheric science community. To this end, it should be possible to find the basic numbers of characteristic atmospheric parameters for the CAO-PDD network of stations already inside this article (without extensive literature search).

You may therefore expand tables 1-3 to include e.g. mean values, trends/tendencies and seasonalities from the individual observations or supply this info by adding representative data sets to Figs. 8. Combine several measurands in the figures. Given the details of the instrument descriptions (370 lines), the corresponding results often stay unnecessarily vague (l 582, l 590, l 616ff, l 630,..) - covering only 227 lines. For example the article does not contain any value for basic aerosol parameters like number- or mass-concentration, absorption- or scattering coefficients and composition).

With these revisions I recommend publication in AMT.

Special comments:

Instrumental part: Though proper operation, calibration and traceability is guaranteed by EUSAAR and ACTRIS conformal sampling and audits, I miss specific information about dry/humid sampling of aerosols by the specific instruments.

L 247ff: Is the nephelometer sample dried? Why do you call this diffusion coefficient and distinguish it from the scattering coefficient? Also 'simple diffusion albedo' sounds 'very French' → single scattering albedo

L 336ff: The Picarro analyser. . . It seems that part of this sentence is missing.

L 360ff: Which consequences has the (commonly executed) change from molybdenum towards blue-light-converter for the consistency of the time series and the long-term trend at PDD?

L 614ff: Could you add some numbers for the inorganic aerosol (anions/cations) concentrations and fractionation?

Fig 4: Is the linear model the appropriate approximation to the observed data over 140 years? Is there no change in the trend during the last decades?

Fig. 8: If these were easily to extract from your database: Could you show several more quantities as box-and-whisker-plots or superimposed as monthly means with percentiles?

Please also note the supplement to this comment:
https://www.atmos-meas-tech-discuss.net/amt-2019-383/amt-2019-383-RC2-supplement.pdf

---

## Author Comment (AC1) · 7 Apr 2020

The paper summarizes long term measurements at a cluster of sites around the puy de Dome near Clermont Ferrand in Central France. It describes the geography of the site, the (history of) deployments of various instruments and the measurement techniques used. It synthesizes major results obtained from the observations and places them into the context of other EU sites. In my mind, the paper is interesting to publish in AMT, because it gives a good review of observations available at this internationally recognized site. This will be useful for the AMT readership. Before publication, the authors should take into account several general and specific remarks:

We thank the reviewer for this positive review and we will provide a revised version of the paper taking into account the suggestions.

1) The data availability is generally not discussed. This is not in line with current practice, and it is even more required for this review paper which has among its aims to better document the available data sets for a more widespread use. The procedure to download available ta sets has to be given.

Yes, we agree with reviewer that data availability should be mentioned. Now, in the revised version, the data availability is described in a new appendix A.

2) A section should be added which shows the type of encountered air masses at the sites, and in particular the PUY site. What are the geographic origins, which part is free tropospheric and from the boundary layer. How does local pollution from Clermont Ferrand affect the site. Are there any small scale meteorological studies addressing this question? Part of information is given in the results sections, but should be grouped in a section prior to it.

Yes, the geographical context was discussed partially in the introduction and result sections. For more clarity, the type of encountered air masses is now discussed in a new appendix C.

3) For all instruments and geophysical variables, information about measurement periods and frequencies, detection limits and uncertainties should be given in synthetic tables.

Tables 1 to 3 have been completed with measurement periods and typical frequencies. The presentation of parameters such as detection limits or uncertainties is very specific for each instrument, and it is difficult to present them in a synthetic homogeneous way in a table. However, clarifications have been made in the text when possible and available to respond to the reviewer's comment.

4) Carefull rereading of the paper is still required. I have corrected several grammer errors, but surely not all.

5) Specific remarks :
P6li190 " Technical description of observation systems " At the beginning of this paper I would like to see a small overview of the measurement gements, roughly corresponding to the titles of the sub-sections.

A small overview sentence has been added.

P6li193 " After preliminary : : :. " In this paragraphe, it seems implicitly that aerosol is mainly sampled in cloudy air masses. The sampling strategy needs further explanation.

We agree with the reviewer that the sentence starting at L193 was somewhat misleading; so we have slightly modified it as following: "Specific inlets for aerosol particles sampling also under cloudy conditions have been deployed

P8Li255 Are there calibration procedures for ACSM or OC/EC ?

Yes, additional text is now included in the section 5.1.1 (chemical properties)
"The instrument response is determined through calibration with 300 nm ammonium nitrate particles. These aerosols are generated from solutions of 0.005 M ammonium nitrate. Aerosols are atomized and then dried to a humidity of < 30% before passing into a differential mobility analyzer to select the size (Freney et al., 2019)."

"Both the ToF-ACSM and the offline filter (OC/EC) analysis methods are regularly checked through calibration and intercomparison at the European center of aerosol calibration (https://www.actris-ecac.eu/), specifically at the aerosol chemical monitor calibration center for the ToF-ACSM (Freney et al., 2019; Crenn et al 2015) and at the Joint Research Center (JRC), Ispra for OC/EC (Cavalli et al., 2010)".

P12l370 VOC measurements What are the storage times for the cartridges, and what are the precautions for sample stability?

Storage times for the cartridges is less than 3 months before analysis by GC-MS. Cartridges have Swagelok caps, and are kept in a dark and cool room (~20 °C) at the laboratory. Storage description has been added in the text.

P18li561 6.1 Trace gases I think, it would be better to regroup the GHG measurements mentioned in the beginning and the end of the section. For each result on tendencies and

seasonal variations, it would be interesting to state the particularity of results at PUY (if there ary any) with respect to other European sites. The impact of local and remote sources on trends and variability should be stated, this is related to the height of the boundary layer. For which source area is this receptor site representative?

As suggested we have grouped the two parts of this section concerning GHG, and we have added sentences in section 6.1 on the representativeness of the measurements and on the relationship between local/remote sources impact and the boundary layer height.

"According to Lopez et al., 2015, the measurements observed at PUY during the night are representative of the central part of France, mostly west of the station. Similarly to other mountain sites like Schauinsland or Monte Cimone, the daytime values are more influenced by local sources, and therefore they are generally excluded in the large scale atmospheric inversions (Broquet et al., 2013; Bergamaschi et al., 2017)."

And

"Due to the diurnal and seasonal cycles of the boundary layer height, compounds influenced by anthropogenic sources are more concentrated in summer than in winter when PUY is in free troposphere. Local and regional emissions will influence the trace gas concentrations especially in summer and daytime."

It is to be noted that a comparison with other Europeans sites has also been mentioned in section 6.2 concerning aerosol particle concentrations.

P18, li574 " : : :., while its primary precursors like NOx has significantly decreased (not shown). " This implies that the seasonal variation should become less marked over time, due to less photochemistry in summer and less titration in winter. Is that what the authors want to state? Can this be derived from figure 8 (from visual quick inspection: might be true, but not sure whether statistically significant?

As well as changes in long-range transport, a reduced titration by NO due to less NOx availability and higher biogenic emissions in a warming climate could explain these trends. This is now précised in the text.

"This feature is consistent with the ozone trend reported at surface stations within the EMEP network between 1990 and 2012." What feature is meant here?

This feature means the slight decrease of ozone observed at PUY and over Europe the last 20 years. The sentence has been rephrased.

P18li583 " The analysis of the 3-years atmospheric time series revealed how the planetary boundary layer height drives the concentrations observed at PUY. Radionuclide measurements are used to determine the boundary layer / free tropospheric conditions.
The CO2 surface flux are estimated and revealed a clear seasonalt cycle, under the influence of plant assimilation, and burning of fossil fuel. "
For these " one sentence statements " about results, are there publications and reports available ?

The references (Farah et al., 2018, and Lopez et al., 2015) have been added.

P19Li594 : " The variability of atmospheric aerosol at PUY shows a marked seasonal variation with a maximum during the summer, " Could the authors be more precise on the aerosol property concerned : PM mass, of which diameter, specific compounds, or particle number ?

The paragraph dedicated to the description of aerosol properties was overall revised to include additional information regarding the variables of interest, including their current levels and, when available, the trends observed over the past years. In particular, the first part of Section 6.2 is about > 10 nm particles number concentration.

P19Li594 : " as well as a non-negligible impact of nucleation (or new particle formation (NPF) events) on the particle size distribution " This argument should be apart from the previous one, probably about aerosol mass.

It is now clearly mentioned that the discussion starting at L590 concerns the particle number concentration, so the connection between the abovementioned arguments should now make more sense.

Page19li604 : "coupled with the atmospheric dynamics captured by the LIDAR located at Cézeaux " I think the above peri-phrase can be deleted, because there is no clue about what lidar measurements contributed to the analysis. Else, please explain more the of lidar measurements.

The brief description of the results reported by Boulon et al. (2011) was slightly rephrased. We hope that the contribution of LIDAR data to this work is now more obvious:
"Size distribution measurements performed simultaneously at PUY and OPME show that for over 45% of the time, NPF events are occurring at high altitude while not occurring at low altitude. Such situation is mostly observed when the planetary boundary layer height derived from LIDAR measurements performed at Cézeaux indicates that the PUY station is close or within the lower FT. The remaining observation show that NPF occurs over the entire atmospheric boundary layer (Boulon et al., 2011)."

P20li631 : " In particular, the general trend is a significant decrease of the daytime aerosol diffusion coefficient during summer and autumn (Pandolfi et al., 2018). " It is not clear in this sentence, that a long term evolution is meant. What is the time scale ?

It is now clearly stated that the observations reported by Pandolfi et al. (2018) were derived from the period 2007-2014. Complementary results regarding the trends of both absorption and scattering coefficients recently reported by Collaud Coen et al. (2020) are in addition mentioned in the revised version of the manuscript.

P20li634 : " This trend is in line with aerosol mass (PM) decay observed at European level, in relation to SO2 emission regulations. " What is then the contribution of sulfate to PM ?

The formulation of this sentence was not clear. There is no direct link demonstrated between the aerosol mass and aerosol coefficient decay. This part of the text has been rephrased.

P20li634 : " and higher elevation sites or ocean-influenced air masses contain a larger fraction of highly hygroscopic aerosols (sea salts). " This is not clear, so for PUY is the larger hygroscopicity related to oceanic air mass origin.

This sentence was in fact misleading, and was thus changed to: "This is reflected by the observations performed at PUY, where ~ 45% of the sampled air masses originate from oceanic regions and tend to contain more hygroscopic particles compared to other sectors (Holmgren et al., 2014)".

P22LI689 : " This long term observation of cloud biological composition is a unique worldwide database. " Where are these data (and others) available ?

The access to the data (including cloud biological composition) is given in Appendix A.
The direct link to cloud biological composition data is http://wwwobs.univ-bpclermont.fr/SO/beam/data.php

P23LI736 : "that the urban layer " the presence of such a layer should be explained much earlier, in a section about the air mass origin of the observations.

In this specific study (Van Baelen and Penide, 2009), the urban layer is defined by the layer between the two different altitude sites Opme and Cézeaux. This has been précised in the text. No specific study has demonstrated a link with local encountered air masses.

P24Ll764 : "atmospheric compartments in order to identify the dynamical links and the strength of their exchanges " Are there Rn measurements that could be exploited to detect the PBL origin of air masses ?

Yes, Rn measurements are used to detect the PBL origin of air masses in the study of Farah et al., (2018).

P24Ll780 : "characterize reactive ($NO_3^-$, $SO_4^{2-}$, $NH_4^+$ and $K^+$) : : :. Species transfer " Are there systematic measurments of rain water composition, which allow documenting their long term evolution ?

Measurements of rain water compositions have been made during specific campaigns, but not systematically over long term. This is now specified in the text.

P24Ll783 : "we observed a seasonality of the washout ratio for radionuclides, with higher value in winter and lower value in summer. " A possible reason for this ?

This is mostly valid for $^7Be$. This observation may ensue from one of the following possible reasons or both of them.
In 2007 precipitation amounts were lower in the winter season. Lower precipitation favored the scavenging yield of every rain event. Conversely, large amounts of precipitation often results from more or less prolonged rain events which do not scavenge anymore the atmosphere after the first minutes or ten minutes. As a result, the probability for a given air mass to have been previously scavenged is less important in winter thus rainwater samples are not « diluted » by clear rain. In the meantime airborne $^7Be$ levels are lower in winter because of the reduction of STE and the lower production of $^7Be$, and in spite of possible concentration increases due to thermal inversion layers. We found the seasonal variation of the washout ratio for $^7Be$ exhibits a similar pattern to that for $SO_4^{2-}$ and monthly means and standard deviations in the same range. This confirms the link between both compounds. Indeed, soon after its production, $^7Be$ become attached primarily to submicron aerosols mostly composed of sulfates (Pinero-Garcia and Ferro-Garcia, 2013).

P37-40, tables 1,2,3 : " Laboratory " It should be something like " if other than LAMP or OPGG or so. The tables should contain in addition more some more specifications: time periods and frequency of measurements (not just starting date), detection limit and uncertainty, site where data base is available.

Yes, this has been previously addressed (main point n°3).

---

## Author Comment (AC2) · 7 Apr 2020

Cézeaux-Aulnat-Opme-Puy De Dôme: A multi-site for the long term survey of the tropospheric composition and climate change

The article summarizes instrumentation, research and conceptual facility design built up at and around the Puy-du-Dome, central France, over the last decades, with historical roots reaching back to more than a century ago. Development, operations and future perspectives of an excellently equipped and integrated observatory are described and main findings shortly pointed out or cited, but often only mentioned. The site's relevance emerges from its frequent use by the scientific community for process dedicated campaigns, field deployment of new measurement techniques and strategies as well as its and the contributing institutions' important roles in national and international research infrastructures.

The description of the CO-PDD observatories, their relevance, concepts, aims and integration is comprehensive and useful to AMT readers. A broad selection of results, including many references, convinces that excellent scientific results have been inferred from the CO-PDD measurements. A review like this may, for brevity and clarity, discuss part of the results qualitatively, however, its added value develops from their combination and their synopsis. I'm missing a number of important figures either in the text, as table(s) or as plots in order to serve as a 'first stop' also for external readers aside the European atmospheric science community. To this end, it should be possible to find the basic numbers of characteristic atmospheric parameters for the CO-PDD network of stations already inside this article (without extensive literature search). You may therefore expand tables 1-3 to include e.g. mean values, trends/tendencies and seasonalities from the individual observations or supply this info by adding representative data sets to Figs. 8.

We think that it is difficult to include this specific scientific information in Tables 1 to 3 that are devoted to technical information. We prefer to discuss these features in the scientific section (6) and to present them in Figures 7 to 9.

Combine several measurements in the figures. Given the details of the instrument descriptions (370 lines), the corresponding results often stay unnecessarily vague (l 582, l 590, l 616, l 630,..) - covering only 227 lines. For example the article does not contain any value for basic aerosol parameters like number- or mass-concentration, absorption- or scattering coefficients and composition).

A new figure (Fig. 9) presenting long series of aerosol particles number has been added and discussed in the section 6.2 :

"The total particle number concentration (> 10 nm) currently measured at PUY is on average ~ $2 \times 10^3$ cm$^{-3}$, which corresponds to intermediate values compared to observations reported from neighboring mountain stations in Europe (Laj et al., 2020), such as for instance Montseny (Spain, 700 m a.s.l., ~ $3 \times 10^3$ cm$^{-3}$) or Jungfraujoch (Switzerland, 3578 m a.s.l, ~ $2 \times 10^2$ cm$^{-3}$). As illustrated on Figure 9, the aerosol number concentration tends to overall

exhibit a slight decrease over the past 15 years at PUY, in the order of -9 ± 5 ×10² cm$^{-3}$/decade. Deeper investigation of this trend is currently performed and will include a more detailed discussion of these aspects."

The section 6.3 has also been completed.

"A median scattering coefficient of ~ 10 Mm$^{-1}$, in the range of values observed at other mountain sites, was obtained by Pandolfi et al. (2018) for the period 2007-2014 at PUY. Seasonal medians in the range 0.7 – 9 Mm$^{-1}$ were in addition more recently reported by Laj et al. (2020) for the year 2017, together with median absorption coefficients of 0.92 and 0.44 Mm$^{-1}$ for spring and autumn, respectively."

With these revisions I recommend publication in AMT.

Special comments:
Instrumental part: Though proper operation, calibration and traceability is guaranteed by EUSAAR and ACTRIS conformal sampling and audits, I miss specific information about dry/humid sampling of aerosols by the specific instruments.

Additional information is now provided at the beginning of section 5.1.1, in connection with the description of WAI: "With the exception of the (N)AIS and PSM, all the instruments described here are operated behind a WAI, in which the aerosol is dried due to the temperature difference between external and internal conditions . In contrast, (N)AIS and PSM, which are further described below, are dedicated to the monitoring of newly formed aerosol particles with diameters less than 10 nm, and are thus located on the roof of the station where they sample through a shorter inlet (~30 cm, non-heated) to limit diffusion losses."

L 247: Is the nephelometer sample dried? Why do you call this diffusion coefficient and distinguish it from the scattering coefficient? Also 'simple diffusion albedo' sounds 'very French' ! single scattering albedo

The nephelometer is operated in the station, behind a WAI, so, same as for all other instruments operated behind the WAI, the sample is dried in the WAI due to the temperature difference between external and internal conditions. We hope that our answer to the previous comment will help clarifying this aspect.
The use of "diffusion" was in fact an obvious French mistake and was banished from the revised version of manuscript!

L 336: The Picarro analyser: : : It seems that part of this sentence is missing.

The sentence has been modified.

L 360: Which consequences has the (commonly executed) change from molybdenum towards blue-light-converter for the consistency of the time series and the long-term trend at PDD?

As the molybdenum converter convert other nitrogen species ($NO_y$) than $NO_2$, the change from molybdenum towards blue-light-converter has an impact on measurement, and the $NO_2$ measured after this technical modification is "real" $NO_2$. Therefore the measured concentration was overestimated for $NO_2$ before December 2012. Thus, the trends have been discussed separately for the period 2003-2012 and 2012-present. We add a sentence to precise this point in the article.

L 614ff: Could you add some numbers for the inorganic aerosol (anions/cations) concentrations and fractionation?

Additional text has been included.
"Aerosol chemical composition monitored at PUY is highly variable but average concentration monitored at PUY over the period April 2015 – February 2016 exhibits the following values: organic 57% (2 µgm-3), followed by sulphate 16% (0.4 $µgm^{-3}$), nitrate 12% (0.3 $µgm^{-3}$), ammonium 10% (0.24 $µgm^{-3}$) and BC 5% (0.13 $µgm^{-3}$) (Farah et al., 2020)."

Fig 4: Is the linear model the appropriate approximation to the observed data over 140 years? Is there no change in the trend during the last decades?

The reviewer is right, the linear model is probably not appropriate for the best estimation of the temperature trend on 140 years. As shown on the figure below, each dataset can exhibit different linear trends. The objective of Fig. 4 is only to give a global overview of the long term temperature measurement including the historical period, but a study dedicated to long term trend estimation analysis could be interesting to perform.

[Figure]

Fig. 8: If these were easily to extract from your database: Could you show several more quantities as box-and-whisker-plots or superimposed as monthly means with percentiles?

This is an interesting suggestion, but we did not successfully superimpose box-and whisker plots on Figures 7 to 9 without losing readability. So we prefer to keep these figures as they are.

---

## Author Response (AR2)

Dear editor,

We thank you for this feedback and we submit a new revised version with the following modifications.

*Line 616 : « This is consistent with SO2 trends »*
*Please put 2 in SO2 as a subscript, please verify also elesewhere.*

OK, this has been done (line 601 in this new version)

*Line 716 : I would put « Clouds » as ageneric form.*

The title of subsection 6.3 is now "Clouds" ("Cloud" in the precedent version). Please tell us if we did not well understand this request.

*Line 1488 : analysis shown that => analysis HAS shown that*

OK, this has been corrected (line 1454 in this new version)

*Line 1525 : Figure C.1: « Logarithm of the number of back-trajectories points arriving at the summit of the PUY station for the 2015-2016 whole period. The white circle has a radius of 20 km and illustrates the category "Near" of Table C1, and the black lines separates the different sectors. »*
*black lines separate without « s »*

OK, this has been corrected (line 1491 in this new version)

*the logarithmic scale is not entirely clear . Is it decadal or natural logarithme ? In the first case , on would have 0 exp10 data points. How many data points are there in total.*

We used the matlab log function which is the natural logarithm.

The total number of back-trajectory points is $6.3 \ 10^6$ and the largest number of points in a 0.5° square is $4.2 \ 10^5$. The upper limit of the colorbar (red) is fixed with this last value.

The caption has been completed with this information.

$\log(4.2 \ 10^5) = 12.937$, and . $\exp(12.937) = 4.4 \ 10^5$

Thank you again for your work to process this paper. We are very pleased with this happy end after a long process!

Best regards,

Jean-Luc Baray and Laurent Deguillaume